# Trained immunity of intestinal tuft cells during infancy enhances host defense against enteroviral infections in mice

Deyan Chen [1,2,9], Jing Wu [2,9], Fang Zhang [3,9], Ruining Lyu [2], Qiao You [2], Yajie Qian[4], Yurong Cai [5], Xiaoyan Tian[2], Hongji Tao[2], Yating He[2], Waqas Nawaz[6] & Zhiwei Wu [2,7,8 ✉]

## Abstract

Innate immune cells have been acknowledged as trainable in recent years. While intestinal tuft cells are recognized for their crucial roles in the host defense against intestinal pathogens, there remains uncertainty regarding their trainability. Enterovirus 71 (EV71), a prevalent enterovirus that primarily infects children but rarely infects adults. At present, there is a significant expansion of intestinal tuft cells in the EV71-infected mouse model, which is associated with EV71-induced interleukin-25 (IL-25) production. Further, we found that IL-25 pre-treatment at 2 weeks old mouse enabled tuft cells to acquire immune memory. This was evidenced by the rapid expansion and stronger response of IL-25-trained tuft cells in response to EV71 infection at 6 weeks old, surpassing the reactivity of naïve tuft cells in mice without IL-25-trained progress. Interestingly, IL-25-trained intestinal tuft cells exhibit anti-enteroviral effect via producing a higher level of IL-25. Mechanically, IL-25 treatment upregulates spermidine/spermine acetyl-transferase enzyme (SAT1) expression, mediates intracellular polyamine deficiency, further inhibits enterovirus replication. In summary, tuft cells can be trained by IL-25, which supports faster and higher level IL-25 production in response to EV71 infection and further exhibits anti-enteroviral effect via SAT1-mediated intracellular polyamine deficiency. Given that IL-25 can be induced by multiple gut microbes during human growth and development, including shifts in gut flora abundance, which may partially explain the different susceptibility to enteroviral infections between adults and children.

**Keywords** Trained Immunity; Tuft Cells; IL-25; SAT1; Anti-Enteroviral Response
**Subject Categories** Digestive System; Immunology; Microbiology, Virology & Host Pathogen Interaction

## Introduction

Innate and adaptive immune systems are typically distinguished by their pathogen specificity and memory capacity (Netea et al, 2019; Ter Steeg et al, 2021). In recent years, the concept has been updated: the innate immune system is also capable of building memory capacity after a primary stimulation and initiating a stronger response during a secondary challenge, known as trained immunity (Jentho et al, 2021). Until now, increasing innate cell types have been shown to acquire immune training. Innate immune cells can be trained by specific cytokines or pathogens, which in turn trigger a cascade of increased robust events, such as enhanced cytokines production or accelerated expansion of trained cells. For instance, natural killer (NK) cells, group 2 innate lymphoid cells (ILC2), and group 3 innate lymphoid cells (ILC3) have been shown to exhibit the trained innate immune phenotype to fight against pathogen infections, which are dependent on immune training with specific cytokines (IL-12 and IL-18, IL-33), and pathogens (*C. rodentium*), respectively (Kleinnijenhuis et al, 2014; Martinez-Gonzalez et al, 2016; Serafini et al, 2022). The above evidence implied that innate immune cells could be trained and gain memory. Therefore, exploring the training capacity of innate immune cells is essential for understanding the interaction between host and pathogens, and developing the treatment of pathogenic infections.

Tuft cells have been identified as innate immune cells in intestinal and lung tissues (Haber et al, 2017). Tuft cells have been reported to undergo rapid and transient hyperplasia after stimulation, who played an important role in the clearance of parasites and bacteria (Xiong et al, 2022; Zhao et al, 2022). However, it is not well understood whether tuft cells can be trained. Actually, tuft cells comprise approximately 0.5% of small intestinal epithelial cells and colonic epithelial cells, but tuft cells are more prevalent in the distal than the proximal small intestine, especially in the ileum. Tuft cells have been reported as originating from Lgr5+ stem cells, and doublecortin-like kinase 1 (DCLK1) or/and transient receptor potential melastatin 5 channel (TRPM5) are identified as tuft cell biomarkers (Chandrakesan et al, 2015; Perniss et al, 2023; Yi et al, 2019). Recent studies demonstrated that intestinal tuft cells expand

[1]Anhui Key Laboratory of Infection and Immunity, Bengbu Medical University, Bengbu, China. [2]Medical School of Nanjing University, Nanjing, Jiangsu, China. [3]Department of Burn and Plastic Surgery, Affiliated Hospital of Zunyi Medical University, Zunyi, Guizhou, China. [4]Nanjing Stomatological Hospital, Medical School of Nanjing University, Nanjing, Jiangsu, China. [5]School of Life Science, Ningxia University, Yinchuan, China. [6]Hôpital Maisonneuve-Rosemont, School of medicine, University of Montreal, Montreal, Canada. [7]State Key Laboratory of Analytical Chemistry for Life Science, Nanjing University, Nanjing, Jiangsu, China. [8]Yunnan Provincial Key Laboratory of Entomological Biopharmaceutical R&D, College of Pharmacy, Dali University, Dali, Yunnan, China. [9]These authors contributed equally: Deyan Chen, Jing Wu, Fang Zhang. ✉E-mail: wzhw@nju.edu.cn

rapidly in response to enteric pathogens infection, which is associated with infection-induced interleukin-25 (IL-25) expression. Interestingly, IL-25 can be upregulated by multiple pathogens infection even when alterations occur in the abundance of intestinal commensal bacteria (Jan et al, 2021; Liu et al, 2023; Zaph et al, 2008). Therefore, IL-25 has been identified as an inducer of tuft cell expansion during microbes infection/colonization. However, the role of IL-25 in intestinal tuft cells training to fight with infection is intriguing.

Our previous study showed that IL-25/tuft cell axis could be activated by multiple enteroviruses infection, including enterovirus 71 (EV71), coxsackievirus A16 (CVA16), CVB3, and CVB4 (Lyu et al, 2024). Enteroviral infections cause multiple human diseases and present an important public health problem among many countries, especially in the Asia-Pacific region. It is worth noting that more than 20 human enteroviruses have been identified to lead to HFMD (hand, foot, and mouth disease), with EV71 being the primary pathogen causing HFMD (Tan and Chu, 2021). EV71 initially established infection in the digestive tract via the fecal-oral route. EV71 belongs to the *Picornaviridae* family, and the viral genomes contain the conserved 5′UTR and 3′UTR on both sides and an open reading frame (ORF) in the center (Wang et al, 2022). In addition, EV71 has a positive sense single-stranded RNA with 7400 nucleotides, consisting of four structural viral proteins (VP1, VP2, VP3, and VP4) and seven nonstructural proteins (2A, 2B, 2C and 3A, 3B, 3C, 3D) (Swain et al, 2022). Notably, enteroviral infections exhibit strong age-related characteristics. Most severe illnesses caused by enteroviruses occur in children under 5 years old. In contrast, children aged above 5 and adults are rarely reported being infected or infected but without symptoms (Dallari et al, 2021). Although, previous studies reported that the age-dependent susceptibility to enteroviruses might be related to the differences in the immature adaptive immune system between infants and adults (Lockhart et al, 2022), the roles of the innate immune system, especially IL-25 mediated tuft cells training, playing in this difference were largely unknown.

In this study, we established EV71 infection model in C57BL/6J and $IL$-$25^{-/-}$ mice and recognized tuft cells by its specific biomarkers TRPM5 (transient receptor potential melastatin five channel) and DCLK1 (doublecortin-like kinase 1) to investigate the training capacity of IL-25 on tuft cells in intestinal tissues and tried to explain the immune training mechanism of the different susceptibility in enteroviral infections between adults and children.

## Results

### EV71 infection induced tuft cell expansion

AG6 mice have been reported as a EV71 susceptibility model by us (You et al, 2023). Therefore, we initially chose AG6 mouse to investigate the relationship between EV71 infection and intestinal tuft cells expansion. Two weeks old AG6 mice were infected with EV71 ($1.2 \times 10^6$ PFU) by intraperitoneal injection (i.p.) for 14 days (Fig. 1A). We detected the presence of EV71 viral RNA in the duodenum, jejunum, and ileum tissues of EV71-infected AG6 mice on day 7 post-infection and found that EV71 tended to primarily infect the ileum during the initial stages of infection (Fig. 1B). And then we detected EV71 viral protein VP1 not only expressed in the intestinal epithelial

cells, but also expressed in tuft cells (DCLK1$^+$ cells) (Fig. EV1). Furthermore, EV71 infection resulted in a shorter intestinal microvillus in the ileum on 7 dpi (Fig. 1C). A time addition experiment showed that EV71 infection induced a significant tuft cell hyperplasia in the ileum at 3, 5 and peaked at 7 dpi and then returned to a basal level at 14 dpi (Fig. 1D,E). Subsequently, we used TRPM5 (green) and DCLK1 (red) co-marked tuft cells and confirmed that EV71 infection did increase the number of TRPM5$^+$/DCLK1$^+$ tuft cells in the ileum tissues of AG6 mice on 7 dpi (Fig. 1F). A similar phenomenon was also obtained in AG6 mice at 2 weeks old following CVA16 infection that CVA16 infection promoted tuft cell expansion on 7 dpi (Fig. 1G,H). Moreover, we have reported that enteroviruses, including EV71, CVA16, CVB3, and CVB4, could induce intestinal tuft cells expansion in enteroviruses infected C57BL/6J mouse model, which implied that this phenomenon was not specific to AG6 mouse (Lyu et al, 2024). In short, the above observations suggested that enteroviral infections, such as EV71 and CVA16, induced a transient expansion of tuft cells in the ileum of the mouse.

### EV71 infection induced tuft cell expansion via promoting IL-25 production

IL-25 was an important factor to mediate the expansion of tuft cells. Here IL-25 mRNA (Fig. 2A) and IL-25 protein (Fig. 2B) were significantly increased in ileum tissues of EV71-infected AG6 mice at 7 dpi. Subsequently, IL-25 protein was also upregulated in EV71-infected HeLa cells, Caco-2 cells, and HT-29 cells (Fig. 2C–E). We further found that the expression of EV71 viral protein 2B, 3AB, or 3C in HeLa cells induced IL-25 protein upregulation as shown by western blot analysis (Figs. 2F and EV2A) and promoted IL-25 production in HeLa cells by IF analysis (Fig. EV2B,C). The above experiments validated that EV71 infection upregulated IL-25 expression in both in vivo and in vitro models. In order to investigate whether enteroviral-induced tuft cells expansion was dependent on IL-25 production, recombinant mouse IL-25 protein (rmIL-25) was used to stimulate C57BL/6J mouse three times (Fig. 2G), and rmIL-25 notably upregulated the expression of DCLK1 and TRPM5 mRNA and induced tuft cells expansion in the ileum tissues of 2 weeks old C57BL/6J mouse (Fig. 2H–K). Furthermore, compared with EV71-infected *WT* mice, the number of DCLK1$^+$/TRPM5$^+$ tuft cells were not increased in EV71-infected $IL$-$25^{-/-}$ mice (Fig. 2L,M), which suggested that EV71 induced tuft cells expansion was associated with IL-25 production.

### IL-25-trained tuft cells in infancy exhibited a stronger anti-enteroviral response at adult stage

The plasticity of training of tuft cells is unclear, therefore, we tried to use rmIL-25 protein to train tuft cell in 2 weeks old AG6 mice (Fig. 3A), the number of tuft cells and IL-25 mRNA were significantly increased after 1 week (Fig. 3B,C), and then they were reduced to a baseline level in ileum tissues in week 4 (Fig. 3D,E). Subsequently, we selected 2 weeks old AG6 mice to receive or not receive IL-25 training, and then the mice were continued to be infected with EV71 after 4 weeks for 7 days (Fig. 3F). After EV71 infection, the number of DCLK1$^+$/TRPM5$^+$ tuft cells were significantly increased in the ileum tissues of rmIL-25-trained mice (Fig. 3G,H). Compared to PBS-trained mice, IL-25-trained mice exhibited a stronger antiviral effect with a lower

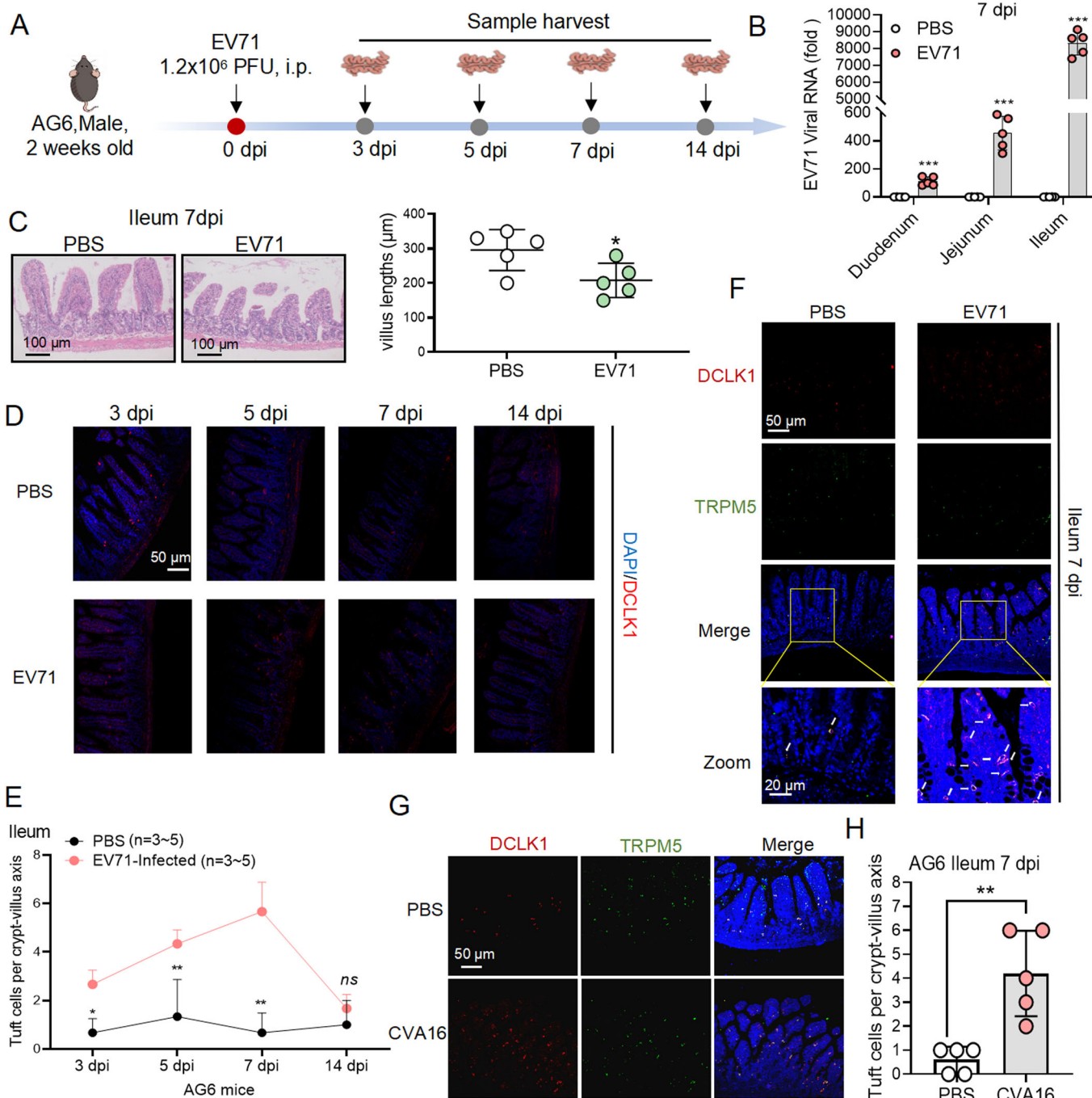

EV71 viral RNA expression in the blood, small intestine (Fig. 3I), and a lower EV71 copies in feces (Fig. 3J). Next, non-lethal titers ($1.2 \times 10^6$ PFU/mL) of EV71 were used to infect 2 weeks old mice, and then re-challenge with EV71 after 4 weeks to further investigate the plasticity of training of tuft cells (Fig. 3K). After EV71 re-challenge, the number of DCLK1$^+$/TRPM5$^+$ tuft cells were significantly increased, which was higher than the PBS-trained group (Fig. 3L,M). An upregulation of DCLK1 and TRPM5 mRNA were consistent with the increased tuft cells expansion (Fig. 3N,O). Furthermore, H&E staining indicated that IL-25-trained mice showed a less disruption on intestinal epithelial barrier integrity

(Fig. 3P). Collectively, intestinal tuft cells could be trained by IL-25 stimulation or EV71 challenge, which exhibited a stronger antiviral effect in trained mice, with the optimal time window for trained immunity being greater than 4 weeks.

## IL-25 as an antiviral effector inhibited enteroviral replication in vitro

The antiviral mechanisms of trained tuft cells are largely unknown. Previous studies showed that tuft cells supported IL-25 production (Varyani et al, 2022). At present, we also observed that IL-25-

**Figure 1.  EV71 infection contributed to tuft cell expansion in mice.**

(A) The schematic diagram for experiments (B–F). Two weeks old AG6 male mice ($n = 14$) were randomly divided into four groups, and they were both infected with EV71 BrCr ($1.2 \times 10^6$ PFU, 100 µL) by i.p. on day 0 and ileum tissues of EV71-infected mice were collected on days 3, 5, 7, and 14 post infection (pi). The control group mice were treated with PBS at the same volume (PBS, 100 µL). The mice were divided into four groups such as 3 dpi ($n = 3$), 5 dpi ($n = 3$), 7 dpi ($n = 5$), and 14 dpi ($n = 3$). (B) qPCR was used to analyze EV71 viral RNA levels in the duodenum, jejunum, and ileum tissues from PBS group ($n = 5$) and EV71-infected group ($n = 5$) on 7 dpi, respectively. (C) H&E staining showed the ileum tissues from PBS-treated and EV71-infected mice. Scale bar = 100 µm. The villus lengths of the ileum from PBS and EV71-infected mice were measured from the left. (D, E) The number of tuft cells was counted in AG6 mice following EV71 infection at different time points. (D) Immunofluorescence (IF) of tuft cell staining in ileum tissues at 3 dpi, 5 dpi, 7 dpi, and 14 dpi from (top) uninfected mice (PBS group) and (bottom) EV71-infected mice, respectively. Scale bar: 50 µm. (E) IF quantification of DCLK1$^+$ tuft cells per crypt/villus at 3, 5, 7, and 14 dpi from (D). The graph showed the distribution of Dclk1$+$ tuft cells in ileum tissues in PBS and EV71-infected mice on days 3, 5, 7, and 14 post infection. Tuft cells were counted in the crypt and villus compartments of $n = 50$ crypt–villus units per mouse with three mice per condition. SD for uninfected mice ($n = 3$–5) and EV71-infected mice ($n = 3$–5). (F) Paraffin-embedded sections of ileum tissues of EV71-infected mice on 7 dpi were stained for tuft cells (DCLK1, red), (TRPM5, green), and cell nuclei (DAPI, blue). Images are representatives of two independent experiments with 3–5 mice. Scale bar: 20 µm. White arrows indicate the co-location of TRPM5 and DCLK1 for tuft cells. (G) AG6 male mice ($n = 10$) at 2 weeks old were randomly divided into two groups ($n = 5$ in each group), and they were infected with CVA16 ($2.0 \times 10^6$ PFU, 100 µL) or PBS (100 µL). After 1 week, ileum tissues were collected, and they were further stained for tuft cells (DCLK1, red; TRPM5, green) and cell nuclei (DAPI, blue). (H) IF quantification of DCLK1$^+$/TRPM5$^+$ tuft cells per crypt/villus at 7 dpi from (G). SD for uninfected mice (PBS group) ($n = 5$) and CVA16-infected mice ($n = 5$). Images are representatives of two independent experiments with five mice in each group. Scale bar: 50 µm. Data were presented as mean ± SD. The major statistical procedures applied were the Shapiro–Wiik test (B, C, E, H), Paired *t*-test (B), Wilcoxon test (E), and student's *t*-test (H). ns not significant; *$P < 0.05$; **$P < 0.01$; ***$P < 0.001$. Source data are available online for this figure.

---

trained mouse exhibited elevated IL-25 mRNA expression in the ileum tissues during EV71 infection (Fig. 4A). Furthermore, both IL-25-trained and EV71-trained mice displayed heightened levels of IL-25 protein in fecal samples as compared to the PBS-trained group following EV71 infection (Fig. 4B). Therefore, we speculated that IL-25 might act as an antiviral effector to inhibit enteroviral replication. In an in vitro assay to evaluate the antiviral effect of IL-25, we chose HeLa cells and Caco-2 cells. HeLa cell line was commonly used in antiviral research since HeLa cells had properties including convenient culture and easy accessibility (Zajac and Crowell, 1969). And Caco-2 cells are a type of human clonal colon adenocarcinoma cells that closely resemble differentiated small intestinal epithelial cells in both structure and function and are commonly utilized in research settings to conduct experiments that simulate the behavior of intestinal epithelial cells in vivo (Anabazhagan et al, 2017; Cortez et al, 2023; Yu et al, 2012). Therefore, HeLa cells and Caco-2 cells were used for assessing the anti-enteroviral effect mediated by IL-25 in vitro in this study. We initially explored the antiviral effect of IL-25 in EV71 infected cells and found that IL-25 overexpression suppressed EV71 viral protein VP1 expression and reduced viral titers by 2 logs in HeLa cells (Fig. 4C,D). Furthermore, rhIL-25 treatment also significantly reduced the expression of EV71 VP1 mRNA and protein at 10 ng/mL (Fig. 4E–G). IL-25 overexpression exhibited a broad-spectrum anti-enteroviral effect, including EV71, CVA16, and CVB3, due to a reduction of 5′UTR mRNA expression and viral titers were observed in IL-25 overexpressed Caco-2 cells (Fig. 4H,I). Moreover, the expression of 5′UTR mRNAs and viral titers of EV71, CVA16, and CVB3 were notably increased in IL-25 knockdown Caco-2 cells (Fig. 4J–L). In summary, the in vitro results suggest that IL-25 may be an antiviral effector of trained intestinal tuft cells, which needs to be further verified by in vivo experiments.

## IL-25 exhibited the antiviral effect in vivo

To further investigate the anti-enteroviral role of IL-25 in vivo, *WT* mice and *IL-25$^{-/-}$* mice at 6 weeks old were both infected with EV71 for 7 days (Fig. 5A). As expected, IL-25 mRNA could not be detected in ileum tissues of EV71-infected *Il-25$^{-/-}$* mouse (Fig. 5B),

and IL-9 mRNA, a downstream cytokine of IL-25, were also detected at low levels in ileum tissues of EV71-infected *IL-25$^{-/-}$* mice (Fig. 5C). Consistent with results in vitro, EV71 viral RNA and viral protein VP1 were higher expressed in ileum tissues of *IL-25$^{-/-}$* mice than *WT* mice (Fig. 5D,E). And EV71-infected *IL-25$^{-/-}$* mice exhibited a disrupted intestinal epithelial barrier integrity in ileum tissues (Fig. EV3). Given that *IL-25$^{-/-}$* mice had no response in tuft cell expansion following EV71 infection, as shown in Fig. 2M, we thought IL-25 deficiency abrogated the intestinal defense of tuft cells-mediated IL-25 production against EV71 infection in mice. Collectively, our in vivo results were consistent with in vitro results that IL-25 negatively regulated EV71 replication.

## Activation of SAT1 contributed to IL-25-mediated antiviral effects

To explore the potential mechanism of IL-25-mediated antiviral effects, we established an *IL-25$^{+/+}$* HeLa cell line via lentiviral transfection assay. And then RNA sequencing (RNA-seq) was conducted to analyze the differential expression of transcriptional genes in *WT* and *IL-25$^{+/+}$* HeLa cells, and SAT1 (spermidine/spermine acetyl-transferase enzyme) mRNA was significantly increased by $\log_2$Fc > 2 in *IL-25$^{+/+}$* HeLa cells (Fig. 6A). SAT1 is the rate-limiting enzyme controlling the first intracellular pathway of polyamine catabolism, and high-level SAT1 expression leads to an overall depletion of intracellular polyamines (Ou et al, 2016). Excitingly, the polyamines are essential for the replication of viruses (Mounce et al, 2017). Therefore, we next focus on the roles of SAT1 expression in IL-25-mediated anti-enteroviral effects. The expression of both SAT1 protein (Fig. 6B) and SAT1 mRNA (Fig. 6C) were upregulated in *IL-25$^{+/+}$* HeLa cells. And rmIL-25 also significantly upregulated SAT1 mRNA expression in ileum tissues of mice (Fig. 6D). Furthermore, SAT1 knockdown (~50%) promoted EV71 replication and the production of its progeny virus in HeLa cells and Caco-2 cells (Fig. 6E–J). And SAT1 overexpression significantly reduced both EV71 replication and the production of its viral progeny in HeLa cells (Fig. 6K,L). In order to investigate the role of SAT1 in IL-25-mdiated anti-enteroviral effects, we further detected EV71-VP1 protein expression and

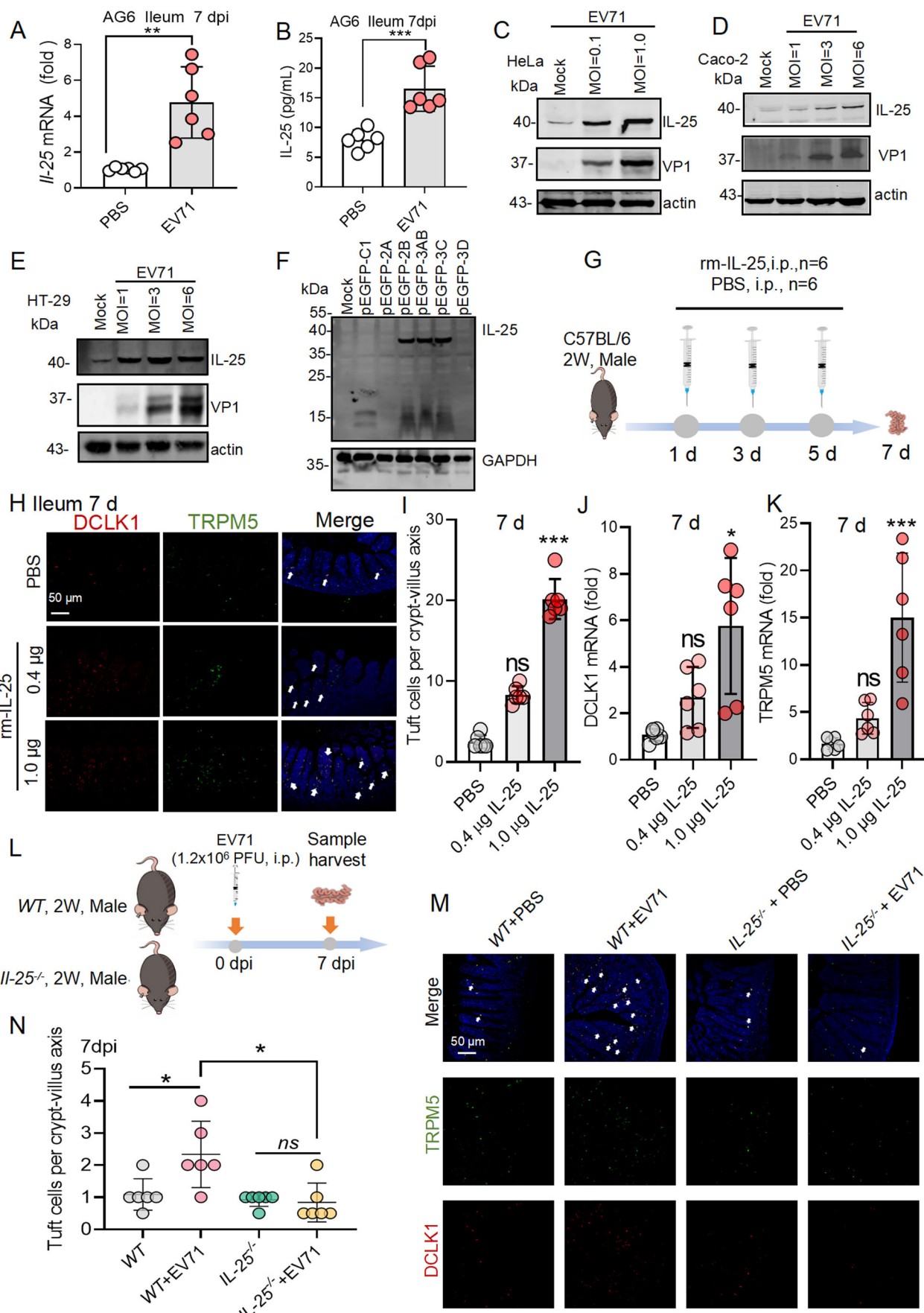

**Figure 2.  EV71 infection increased tuft cell expansion via IL-25 upregulation.**

(A, B) IL-25 mRNA and protein expression were analyzed by qPCR and ELISA in ileum tissues from 2 weeks old AG6 mice (PBS group, $n = 5$) and EV71-infected group (EV71 group, $n = 5$) at 7 dpi as shown in Fig. 1A, respectively. (C–E) HeLa, Caco-2, and HT-29 cells were infected with EV71 at different MOIs (MOI = 0.1 and 1.0 in HeLa cells) or (MOI = 1, 3, 6 in Caco-2 and HT-29 cells) for 24 h and IL-25 protein levels were measured by western blotting analysis. (F) Western blotting detected IL-25 expression in the presence of EV71 2A, 2B, 3AB, 3C, and 3D in 293T cells for 24 h. (G) The experiment design from H to K was shown. C57BL/6J male mice at 2 weeks old were treated with PBS ($n = 6$) and different doses of rmIL-25 0.4 μg ($n = 6$) and 1.0 μg ($n = 6$) on days 1, 3, and 5, respectively. And the ileum tissue samples were collected on day 7. The samples were used for IF staining and qPCR analysis. (H, I) Paraffin-embedded sections of ileum tissues in rmIL-25 protein-treated mice were stained with DCLK1 antibody and TRPM5 antibody to confirm tuft cells (DCLK1, red), (TRPM5, green) and cell nuclei (DAPI, blue). Images are representatives of three independent experiments with six mice. Scale bar: 50 μm. After then, IF quantification of DCLK1$^{+/}$TRPM5$^+$ tuft cells per crypt/villus. (J, K) qPCR analysis was used to detect DCLK1 and TRPM5 mRNA in PBS-treated and rmIL-25-treated ileum tissues on day 7. SEM for PBS-treated mice ($n = 6$) and rmIL-25-treated mice ($n = 6$). (L) The experiment design for M to N was shown. WT and *IL-25*$^{-/-}$ male mice at 2 weeks old were infected with EV71(1.2 × 10$^6$ PFU, 100 μL) by i.p. on day 0 and ileum tissues were collected on 7 dpi. (M) Ileum tissues from PBS- and EV71-infected *WT* and *IL-25*$^{-/-}$ mice were stained for DCLK1-expressing tuft cells and TRPM5-expressing tuft cells. Scale bar: 50 μm. (N) IF quantification of tuft cells in *WT* ($n = 6$) and *IL-25*$^{-/-}$ mice ($n = 6$) with EV71 infection on 7 dpi. The means of villus/crypt ratio of tuft cell numbers are shown. Data expression as Mean ± SD, the major statistical procedures applied were: Shapiro-Wilk test (A, B, I–K, N), *F*-test (I–K, N), Wilcoxon test (A, B), one-way ANOVA and Dunnett's *t*-test (I–K, N), SNK-q test (N). ns no significant difference; *$P < 0.05$, **$p < 0.01$, ***$P < 0.001$; ns not significant ($p > 0.05$). All graphs show the mean ± SD for six individual mice. Source data are available online for this figure.

progeny virus in *Sat1* knockdown *Il-25*$^{+/+}$ HeLa cells. Here, SAT1 was a downstream component of IL-25 since *Sat1* knockdown in HeLa cells abrogated the IL-25-mediated anti-EV71 effect (Fig. 6M,N). In addition, an SAT1-specific inhibitor, diminazene aceturate (DA), canceled the rhIL-25-mediated anti-EV71 effect in HeLa cells (Fig. 6O). Importantly, stably SAT1 knockdown partly canceled rhIL-25-mediated suppression on EV71/CVA16/CVB3 replication and progeny viral production in Caco-2 cells (Fig. 6P,Q). Taken together, IL-25 positively regulated SAT1 expression and IL-25-SAT1 axis had a broad-spectrum antiviral activity against EV71/CVA16/CVB3 via promoting SAT1 upregulation.

## DA countered IL-25-mediated antiviral effects via modulating SAT1 expression in mice

To investigate the role of SAT1 in IL-25-trained tuft cell-mediated anti-enteroviral effects, 2 weeks old AG6 mice were treated with rmIL-25 and grown to 6 weeks old, and then, they were infected with EV71 for 7 dpi with or without the DA treatment, and SAT1-specific inhibitor (Fig. 7A). IL-25-trained mice with DA treatment had a more weight loss following EV71 infection as compared to IL-25-trained mice with Vehicle (Fig. 7B). Moreover, DA significantly reduced IL-25-trained mediated SAT1 mRNA expression (Fig. 7C), suggesting that DA inhibition of SAT1 activity mitigated the IL-25-mediated SAT1 upregulation. As expected, DA treatment notably rescued EV71 viral RNA levels in the ileum, duodenum, jejunum tissues, and blood (Fig. 7D–G). In addition, DA treatment partially reversed viral titers in the feces of IL-25-trained mice (Fig. 7H), suggesting that the anti-enteroviral effects mediated by IL-25-trained tuft cells were dependent on SAT1 upregulation.

## IL-25-induced SAT1 expression might be involved in intracellular polyamine depletion-mediated antiviral effects in vitro

SAT1, a rate-limiting enzyme, was involved in the first intracellular pathway of polyamine depletion, and the polyamines played an essential role in the growth of RNA viruses (Mounce et al, 2016a; Mounce et al, 2016b). Then, we explored whether the IL-25-SAT1 axis was involved in modulating the level of polyamine to affect the

replication of enteroviruses. Thus, we found that rhIL-25 upregulated SAT1 expression, together with a reduction of the polyamine level in Caco-2 cells (Fig. 8A,B); on the contrary, lenti-shSAT1#3 abrogated the effect of rhIL-25-mediated downregulation of polyamine (Fig. 8A,B). Moreover, OD values revealed that rhIL-25-mediated decreased level of polyamine depended on SAT1 expression (Fig. 8C). Therefore, we thought that the IL-25-SAT1 axis had a negative regulation of the polyamine production, which has been reported to be involved in regulating RNA viral replication.

## Discussion

Innate immune cells are the "doorkeepers" of the immune system and play key roles in fighting against pathogenic infections. Tuft cells are a minor type of intestinal epithelial cells, which have been identified as innate immune cells, and play a critical role in resistance to parasitic and bacterial infections (Xiong et al, 2022). Innate immunity is well known for its non-specificity, which includes the activation process of innate immunity and its biological functions. The development of immunological memory is a new biological function of innate immune cells. Therefore, investigating the mechanism of trained immunity is of great importance in exploring new approaches to prevent infectious diseases. Here, we reported that intestinal tuft cells can be trained by rmIL-25 or enteroviral-induced IL-25 in mice in infancy, which supported a stronger anti-enteroviral response in adult mice, with the optimal time window for trained immunity being greater than 4 weeks. Mechanically, trained tuft cells produced more IL-25, which suppressed enteroviral infections via SAT1-mediated intracellular polyamine deficiency.

IL-25 had been identified as an "alarmin" that could be induced by microbial colonization or infection, which implied that IL-25-mediated tuft cell training might occur periodically. For example, IL-25 production may be associated with intestinal commensal bacteria. The host gut microbiota has been established at birth, and the host intestinal systems undergo a dynamic process of gut microbial ecological succession to remodel the gut immune defense from birth to infancy (Liu et al, 2023). And intestinal symbiotic bacteria genera colonization, including Bifidobacterium,

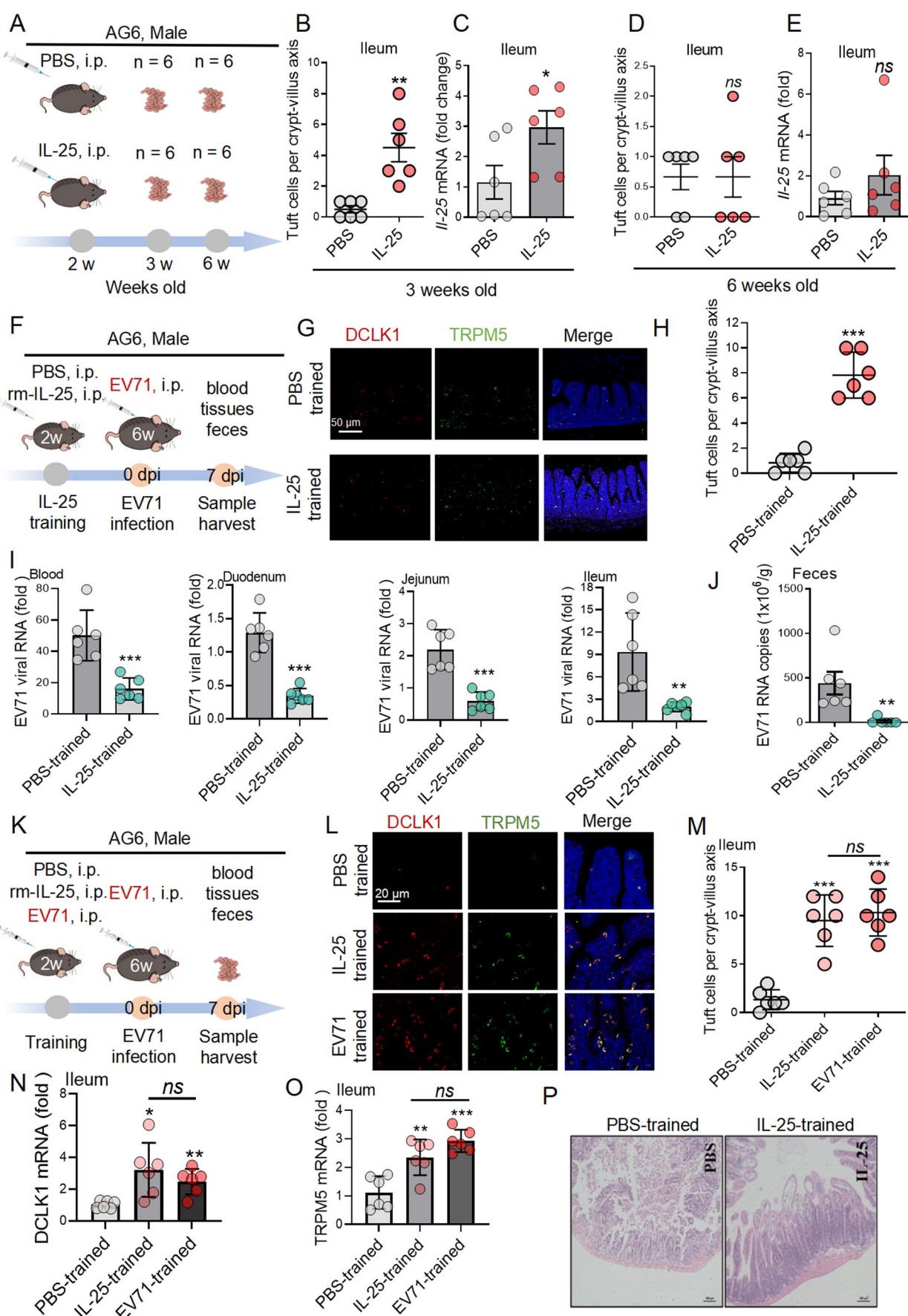

**Figure 3. IL-25-trained tuft cells in infancy had a reduced burden of EV71 in adult mice.**

(A) The experiment design for B–E was shown. AG6 male mice at 2 weeks old in each group ($n = 12$) were treated with PBS (100 µL) and rmIL-25 protein (1 µg, 100 µL) three times, respectively. (B, C) IF quantification of DCLK1+ tuft cells in per crypt/villus and qPCR analysis of il-25 mRNA level from PBS-stimulated and IL-25-stimulated ileum tissues at 3 weeks old ($n = 6$ in each group). (D, E) IF quantification of DCLK1+ tuft cells in per crypt/villus and qPCR analysis of il-25 mRNA level from PBS-stimulated and IL-25-stimulated ileum tissues at 6 weeks old ($n = 6$ in each group). (F) The experiment design for G–J was shown. One group of AG6 male mice ($n = 6$) at 2 weeks old received rmIL-25 protein (1.0 µg, 100 µL, three times) and the other group mice ($n = 6$) received an equal volume of PBS (100 µL, three times) by intraperitoneal injection. After 4 weeks, all of the mice who received PBS or rmIL-25 treatment at 6 weeks old were both infected with EV71 (1.2 × 10^6 PFU, 100 µL) by intraperitoneal injection on day 0. Lastly, three sections of the small intestines, blood, and feces were collected from PBS-trained and IL-25-trained mice on day 7 p.i. (G) IF staining of DCLK1+(red) and TRPM5+(green) tuft cells from PBS-trained and IL-25-trained mice, respectively. Scale bar: 50 µm. (H) IF quantification of DCLK1+/TRPM5+ tuft cells in per crypt/villus as shown in (G) ($n = 6$ in each group). (I, J) qPCR analysis of EV71 viral RNA level from PBS-trained and IL-25-trained mice in the blood, duodenum, jejunum, ileum, and feces, respectively. (K) The experiment design for (L–P) was shown. One group of AG6 male mice ($n = 6$) at 2 weeks old received an equal volume of PBS (100 µL, three times) by intraperitoneal injection; The second group were received rmIL-25 protein (1.0 µg, 100 µL, three times); The third group mice ($n = 6$) were infected with EV71 (1.2 × 10^6 PFU, 100 µL) by intraperitoneal injection. After 4 weeks, all of mice who received PBS, rmIL-25 treatment or EV71 treatment at 6 weeks old were all infected with EV71 (1.2 × 10^6 PFU, 100 µL) by intraperitoneal injection on day 0. Lastly, blood, three sections of the small intestines and feces were collected from PBS-trained, IL-25-trained and EV71-trained mice on 7 dpi. (L) IF staining of DCLK1+ and TRPM5+ tuft cells in ileum tissues from PBS-trained, IL-25-trained and EV71-trained mice, respectively. Scale bar: 20 µm. (M) IF quantification of DCLK1+/TRPM5+ tuft cells in per crypt/villus from (K). SEM for PBS-trained mice ($n = 6$) and IL-25-trained mice ($n = 6$). (N, O) qPCR analysis of DCLK1 and TRPM5 mRNA in ileum tissues from PBS-trained, IL-25-trained and EV71-trained AG6 mice on 7 dpi. **$P < 0.01$ and ***$P < 0.001$ vs. PBS-trained mice. (P) The ileum tissues from PBS-trained and IL-25-trained mice on 7 dpi were fixed and subjected to H&E staining. Scale bar:100 µm. Data expression as Mean ± SD, the major statistical procedures applied were: Shapiro–Wiik test (B–E, H–J, M–O), F-test (M–O), Student t-test (B–E, I, J), one-way ANOVA and SNK-q test (M), Kruskal–Wallis (H) (N, O). ns no significant difference; *$P < 0.05$, **$P < 0.01$, ***$P < 0.001$. Source data are available online for this figure.

Ruminococcus, Methanobrevibacter, and Bilophila were able to induce IL-25 secretion. The IL-25 transcripts in the intestinal cells of mouse under sterile condition were lower than mouse under conventional conditions; and IL-25 was significantly reduced in the colon tissue of mouse with antibiotic treatment (Zaph et al, 2008). In addition, the roles of pathogenic bacteria in the processes of IL-25 training tuft cells should not be ignored. Clostridium difficile (C. difficile), which is a gram-positive bacterium and widely distributed in the intestinal tract of humans, could induce IL-25 expression. Furthermore, pathogens, including parasites, may cooperate to participate in driving IL-25-trained tuft cell expansion. For example, Toxoplasma gondii (T. gondii), affecting 30–50% of children in the world, is a strong inducer of IL-25 (Lima and Lodoen, 2019). Harris NL et al. revealed that parasitic infection, including helminths and arthropod parasites, induced IL-25 upregulation and promoted the expansion of intestinal tuft cells (Zaiss et al, 2013). Marie et al demonstrated that fecal microbiota transplantation increased colonic IL-25 and reduced inflammation in patients with recurrent C. difficile (Jan et al, 2021). Interestingly, IL-25 could also be induced by multiple intestinal food allergens challenge or contaminants exposure; and these allergens and contaminants not only directly induced IL-25 production from the intestinal epithelium, but also possible to indirectly induce IL-25 secretion from the intestinal epithelium via establishing a link through the 'lung-gut' axis (Parrón-Ballesteros et al, 2023). IL-25 is a well-known effective inducer for tuft cells expansion, and we found that intestinal tuft cells could be trained by IL-25. Therefore, we speculated that tuft cells trained by IL-25 is a common phenomenon during human growth and human development. Interestingly, enteroviruses, including EV71, is more preferred to infect children under 5 years of age, and our present results showed that EV71-induced IL-25 production was associated with tuft cell expansion. Therefore, we propose the hypothesis that children under 5 years old are more susceptible to EV71, which may be associated with incomplete intestinal tuft cell immune training, which may not be single-dependent on EV71-related IL-25 production. In summary, our data may partially support the

different susceptibility in EV71 infection between children and adults.

Previous studies have uncovered that tuft cells, an epithelial cell type, are the single source of IL-25 in the small intestine (Kortekaas Krohn et al, 2018). In this study, we found that IL-25-trained tuft cell further supported more IL-25 production. IL-25 has broad-spectrum anti-microbial activity, including bacteria and parasites (Wu et al, 2022), and the enteroviruses we reported in this study. For bacterial infections, IL-25 had been reported to reduce the host mortality and tissue pathology in C. difficile infection; IL-25 had been reported to support the host defense against Staphylococcus aureus (S. aureus) via modulating the immunity. For parasitic infections, IL-25 promotes the clearance of parasites, including worms, T. gondii, plasmodium, etc. (Gerbe et al, 2016). However, the relationship between IL-25 and viral infections is largely unknown. At present, we observed that recombinant IL-25 or IL-25-trained tuft cells-derived IL-25 exhibited a broad-spectrum anti-enteroviral effect, including EV71, CVA16, and CVB3. However, Teresa C Williams et al. found that exogenous IL-25 treatment increased the rhinovirus (RV) load, a common respiratory virus (Williams et al, 2022), which was different from our findings. This may be associated with the different animal models. They found that anti-IL-25 monoclonal antibody (LNR125) treatment increased the RV load in an asthma model established by BALB/c, while they did not observe any changes in coronavirus 229E and RV viral load with LNR125 treatment in bronchial epithelial cells (BECs), which implied that ovalbumin established asthma BALB/c mouse model led to a different role for IL-25 in viral infections. Furthermore, viral specificity should not be ignored due to RV has a robust association with asthma, and IL-25 may support RV replication. In short, we speculated that IL-25-trained tuft cells in infancy supported more IL-25 production to fight with enteroviral infections. Possibly, this IL-25-mediated tuft cell training effects not only protect the host from enteroviral infections in adults, but also extend protection against unrelated pathogens with marked age-susceptibility differences, which requires more research to confirm in the future.

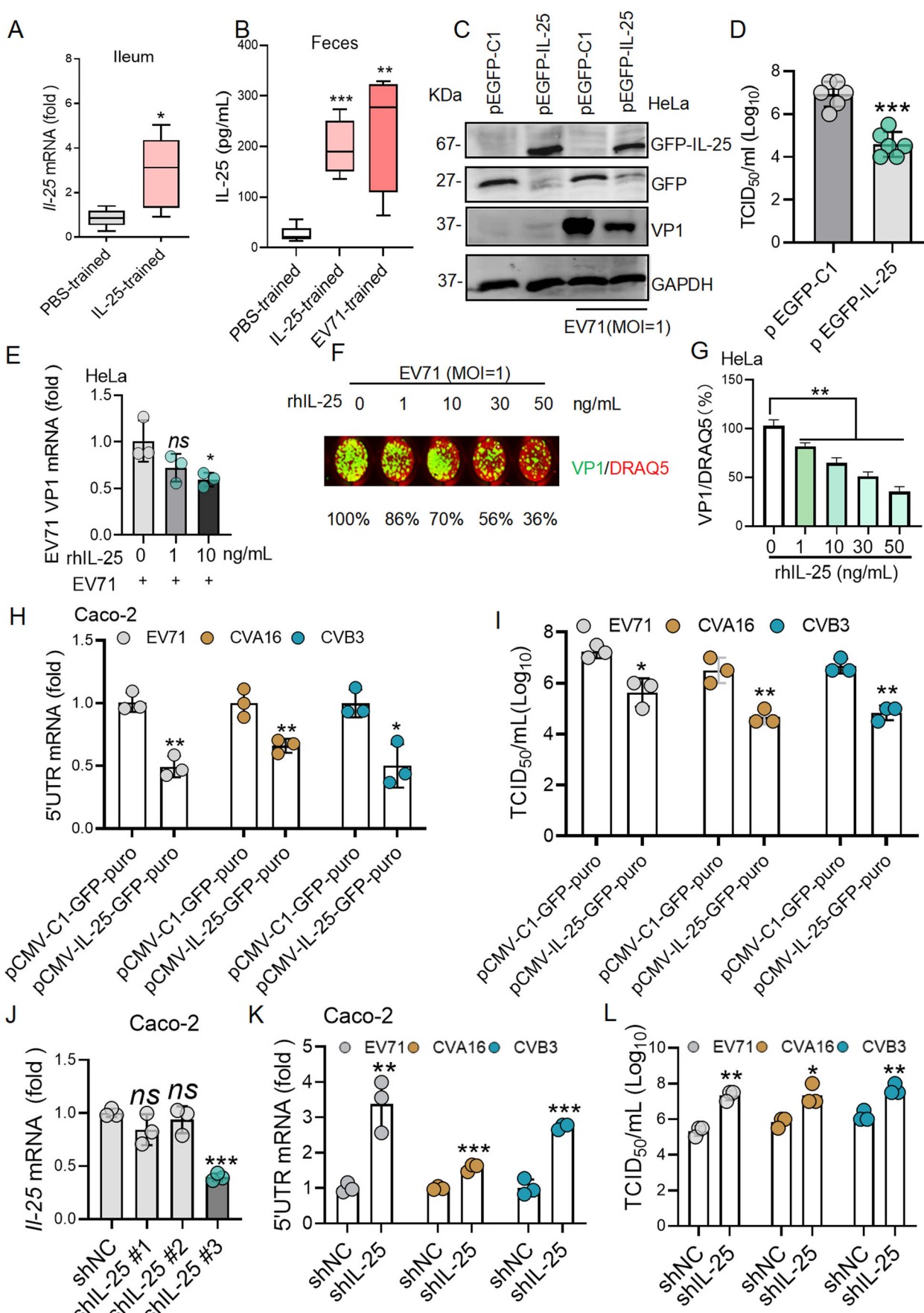

**Figure 4. IL-25 as an antiviral effector inhibited enteroviral replication in vitro and in vivo.**

(A) qPCR analysis of IL-25 mRNA level in ileum tissues from PBS-trained and IL-25-trained mice (*n* = 6 biological replicates in each group) on 7 dpi as shown in Fig. 3F. Box plot shows the quantification of IL-25 mRNA level indicating median, Q1/Q3 and max/min value whiskers of fold-change in ileum tissues from IL-25-trained vs PBS-trained. (B) ELISA assay analysis of IL-25 protein in the feces from PBS-trained, IL-25-trained mice, and EV71-trained on 7 dpi as shown in Fig. 3K (*n* = 6 biological replicates in each group). The box plot shows the level of IL-25 protein indicating median, Q1/Q3, and max/min value whiskers of fold-change in the feces from IL-25-trained or EV71-trained vs PBS-trained. (C) Western blotting analysis was performed to measure EV71 viral protein VP1 in the IL-25 overexpressing group in HeLa cells as compared to the empty Vector group following EV71 infection for 24 h. (D) TCID$_{50}$ assay was used to measure the viral titers in cell supernatant from pVector and pIL-25-HeLa cells. *n* = 6 biological replicates. (E) qPCR analysis was performed to measure VP1 mRNA in the presence of different doses of rhIL-25 (1 ng/mL and 10 ng/mL) for 24 h in HeLa cells. *n* = 3 biological replicates. (F) In-cell western blotting was performed in a 96-well plate to measure VP1 protein expression in the presence of rhIL-25. (G) Amount of VP1 protein relative to DRAQ5 was obtained from (F) (*n* = 3 biological replicates). (H, I) Lenti-IL-25-overexpressing-Caco-2 cells were infected with EV71, CVA16, and CVB3 for 24 h. qPCR analysis of 5′UTR mRNA from EV71, CVA16, and CVB3-infected Caco-2 cells for 24 h (*n* = 3 biological replicates). TCID$_{50}$ assay was used to measure the viral titers in cell supernatant from lenti-pVector and lenti-pIL-25-Caco-2 cells (*n* = 3 biological replicates). (J) qPCR analysis of IL-25 mRNA level on the Caco-2 cells with shIL-25#1, shIL-25#2 and shIL-25#3 as compared to Caco-2 cells with shNC (*n* = 3 biological replicates). (K, L) Caco-2 cells with shIL-25#3 and shNC were infected with EV71, CVA16, and CVB3 for 24 h, respectively. They were analyzed with qPCR and TCID$_{50}$ assay to measure the 5′UTR mRNA (*n* = 3 biological replicates) and viral titers from the cell supernatant (*n* = 3 biological replicates), respectively. Data expression as Mean ± SD, the major statistical procedures applied were: Shapiro–Wiik test (A, B, D, E, G, K, L), *F*-test (B, E, G, J), Wilcoxon test (A), Kruskal–Wallis *H* (B, J), Student's *t*-test (D), one-way ANOVA and Dunnett's *t*-test (E, G), Paired *t*-test (H, I, K, L). **P* < 0.05, ***P* < 0.01 and ****P* < 0.001. Source data are available online for this figure.

The mechanism of IL-25 inhibiting enteroviral replication is intriguing. Then, SAT1, a rate-limiting enzyme, has been reported to join in the first intracellular pathway of polyamine depletion. In our present study, RNA sequencing further indicated that IL-25 positively modulated SAT1 expression to reduce EV71 replication in vitro and in vivo. Our results suggested that SAT1 not only served as a downstream effector of IL-25, but also was involved in depleting the polyamine level because the polyamine played an essential role in the growth of RNA viruses. Furthermore, previous studies have reported that SAT1 activation led to polyamine deletion, which could restrict viral replication (Mounce et al, 2016b). Moreover, SAT1-mediated polyamine metabolism, who facilitated cGAS activation by decreasing the level of spermine and spermidinea and inhibited herpes simplex virus 1 (HSV-1) replication in vivo (Zhao et al, 2023). In our present study, IL-25-SAT1 axis restricted EV71/CVA16/CVB3 replication via upregulating SAT1 upregulation, which gave us a speculation that IL-25-SAT1 axis positively regulating SAT1 expression led to a decreased level of polyamine to suppress viral replication among RNA viruses. In our in vivo assay, SAT1 mRNA significantly upregulated in IL-25-trained ileum tissue than PBS-trained group. Mounce et al, reported IFN-β as an inducer of SAT1 (Mounce et al, 2016b), while we reported IL-25 as another inducer of SAT1. A previous study showed that p53, a transcription factor, promoted SAT1 expression, and further suggested SAT1 was a direct p53 target (Ou et al, 2016). Interestingly, p53 also plays a critical role in the activation of the tuft cell-IL-25-type 2 innate lymphoid cell circuit (Chang et al, 2021). However, whether the relationship between IL-25-SAT1 axis and polyamine deletion has been employed in trained immunity will need to be explored in the future.

Interferons (IFNs) are common and important cytokines that inhibit viral infections. Currently, we are utilizing IFN-α/β/γ receptor-deficient AG6 mice on a C57BL/6J genetic background to study the relationship between enteroviral infections and tuft cell expansion for two main reasons. Firstly, our previous study showed that EV71-Brcr strain will be completely cleared in WT mice on days 5–7 after peaking on day 3 (Lyu et al, 2024), and AG6 mouse allows EV71-Brcr strain to replicate in vivo for 7 days reported by us (You et al, 2023). AG6 mice serve as a robust model for viral infections and can allow viral replication for a long-term. AG6 mouse has been reported as a susceptibility model for multiple viruses, including ZIKA virus, dengue virus, etc. (Liu et al, 2016; Sun et al, 2020). Secondly, AG6 mice have the potential effect to amplify IL-25-tuft cell signaling. IL-25-induced tuft cells expansion may be dependent on IL-25-mediated ILC2 proliferation and ILC2-associated IL-13 production. Han M et al, found that in immature mice, IFN-γ inhibited ILC2 proliferation and IL-13 production (Han et al, 2017). Moreover, multiple studies showed that IFNs were negative regulators of ILC2s. In these studies (Kabata et al, 2018; Kabata et al, 2016; Wang et al, 2023), they suggested that IFNs might be inhibitory factors for tuft cells expansion. At present, AG6 mice lacking the IFN-α/β/γ receptor, potentially amplify IL-25-tuft cell signaling effects. the phenomenon of tuft cell expansion induced by EV71 infection or IL-25 stimulation is not exclusive to AG6 mice, as we have also observed the same response in EV71-infected or IL-25-stimulated C57BL/6 J (WT) mice (Lyu et al, 2024). In conclusion, AG6 mice present a promising model for exploring the interplay between viral infections and tuft cell responses.

Although the trained immunity effect on the tuft cells seems to be robust in our study, the potential training effect of IL-25 on other intestinal innate immune cells, for instance, macrophages or lymphocytes, should not be ruled out. For macrophages, macrophages could retain a long-lasting memory of previous microbial encounters, known as trained innate immunity. This enhanced feature of trained immune macrophages enables them to generate a more robust transcriptional response that is qualitatively and quantitatively distinct from that of untrained cells (Locati et al, 2020). With the knowledge available, macrophages can be trained by LPS, β-glucan, or microbial metabolite long-term exposure (Netea et al, 2016). IL-25 had been identified as an inducer for M2 macrophage polarization (Feng et al, 2018). M2 macrophage polarization can protect the host from a viral infection, including CVB3 (Wang et al, 2017). However, it remains unclear whether IL-25 plays a role in the training of macrophages. For lymphocytes, it has been reported that trained immunity in NK cells are important for its non-specific protection to infection. Bacillus Calmette-Guérin, IL-12, IL-15, IL-18, etc., have been reported to have the capacity of inducing NK cell-trained immunity (Kleinnijenhuis et al, 2014; Zhu et al, 2022). ILCs have been recently discovered as innate immune cells that can be trained (Ham et al, 2022). Until

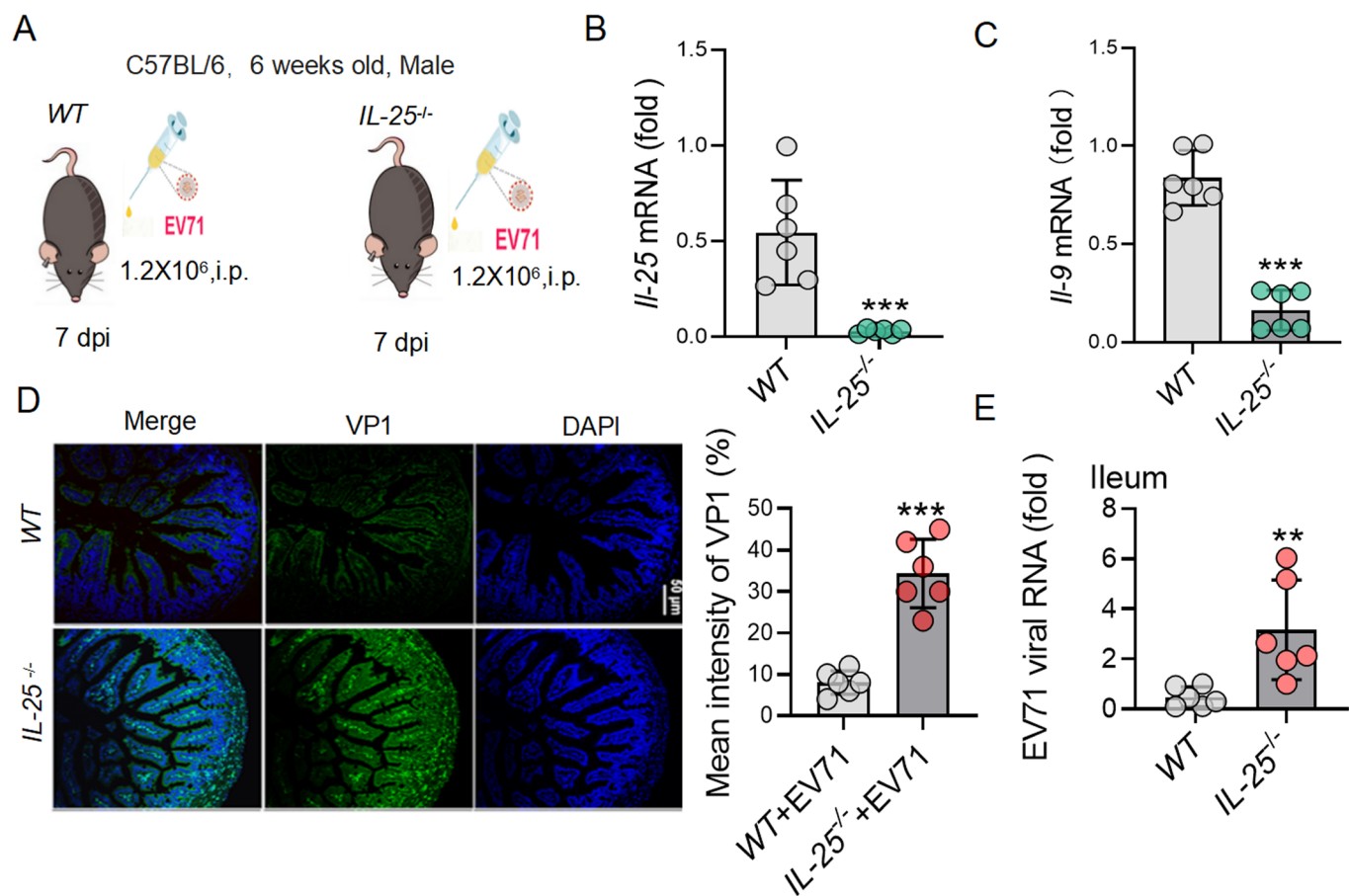

**Figure 5. IL-25 as an antiviral effector inhibited enteroviral replication in vivo.**

(A) The experiment design for (B–E) was shown. *IL-25* $^{-/-}$ mice ($n = 6$) and C57BL/6J wide type (WT) male mice ($n = 6$) at 6 weeks old were both infected with EV71 (1.2 × 10$^6$ PFU, i.p., 100 µL) and ileum tissues were collected at 7 dpi. (B) qPCR analysis of IL-25 mRNA in ileum from *IL-25* $^{-/-}$ mice ($n = 6$) and WT mice ($n = 6$). (C) qPCR analysis of IL-9 mRNA in ileum from *IL-25* $^{-/-}$ mice ($n = 6$) and WT mice ($n = 6$). (D, E) IF staining of EV71 viral protein VP1 and qPCR analysis of EV71 viral RNA level in ileum tissues of EV71-infected *WT* and EV71-infected *IL-25$^{-/-}$* mice, respectively. The fluorescence intensity of VP1 protein was presented as mean ± SD ($n = 6$). ***$P < 0.001$ vs. EV71-infected WT mice. Cell nucleus (DAPI, blue). Scale bar: 50 µm. Data expression as Mean ± SD, the major statistical procedures applied were: Shapiro–Wilk test (B–E), Wilcoxon test (B, C), and Student's *t*-test (D, E). **$P < 0.01$, ***$P < 0.001$. Source data are available online for this figure.

now, ILC1 can be trained by hapten and MCMV-m12; ILC2 can be trained by ASP, PAP, IL-33, and S.venezuelensis; ILC3 can be trained by *C.rodentuim* (Serafini et al, 2022). IL-25 is important for ILCs activation, however, the role of IL-25 playing in lymphocytes is largely unknown. In short, IL-25, as an innate immune molecule, may also induce trained immunity in multiple innate immune cells, therefore, more research needs to be investigated in the future.

However, there are limitations in this study. Based on the current evidence, we have not been able to demonstrate that the increased expression in IL-25 and SAT1 is solely due to the expansion of IL-25-trained tuft cells. Therefore, further studies are needed in the future. Furthermore, it remains unclear whether IL-25-induced tuft cell training occurs directly through the tuft cell or through its progenitor cell. Tuft cell expansion is associated with IL-25 stimulation. Fortunately, it is well established that IL-25-derived tuft cells expansion is

dependent on IL-25-mediated ILC2s activation (von Moltke et al, 2016). IL-25-stimulated ILC2s secreting IL-13 that acts on epithelial crypt progenitors to promote the differentiation of tuft cells, leading to an elevated tuft cell frequencies (von Moltke et al, 2016). Therefore, we speculated that IL-25-trained tuft cells may be associated with ILC2s, but not directly with progenitor cells of tuft cells or tuft cell that specifically drive the secondary response. Nevertheless, further exploration of the specific mechanisms of IL-25-derived tuft cell training is still needed in the future.

Our current study provided a new perspective on intestinal innate immunity during enteroviral infections and offers a potential explanation for the susceptibility to EV71 infection with different ages. The observation may open up a new avenue for exploring a therapeutic strategy to fight against enteroviral replication via manipulating IL-25 production to train tuft cells in infancy.

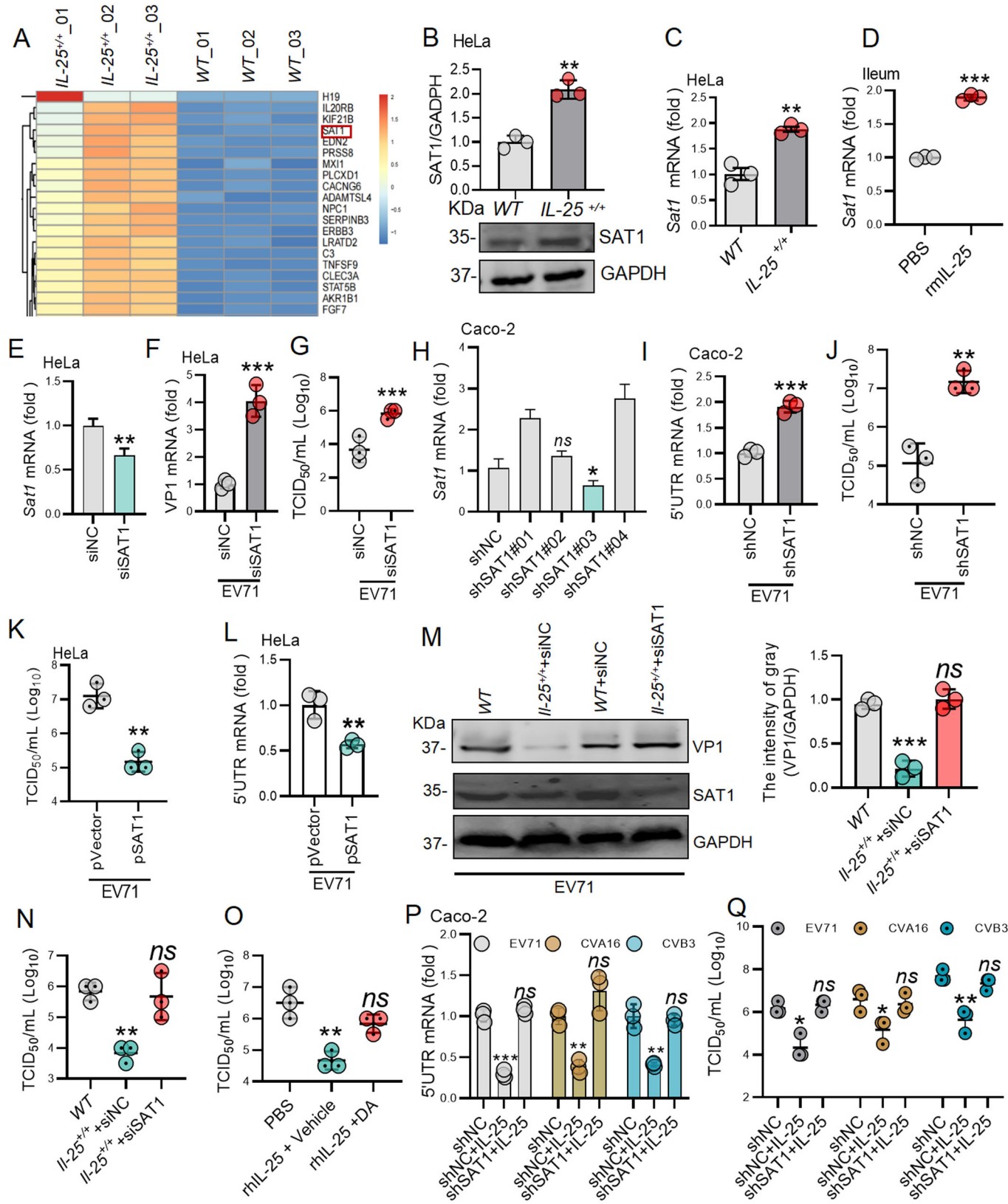

**Figure 6. IL-25-mediated anti-enteroviral effects via upregulating SAT1 expression.**

A heatmap showing the relative expression levels of differentially expressed reported genes was shown. The columns for $IL$-$25^{+/+}$_ HeLa cells or $WT$_ HeLa cells represent the results from three biological replicates. DEGs were selected with fold change >2 and $P$ value <0.05. (B) Western blotting analysis of SAT1 protein in $WT$-HeLa cells and $IL$-$25^{+/+}$-HeLa cells in the absence or in the presence of EV71 infection for 24 h. Quantification of western blotting bands shows the protein level of SAT1 ($n = 3$ biological replicates). (C) qPCR analysis of SAT1 mRNA in $WT$_HeLa cells and $IL$-$25^{+/+}$_HeLa cells ($n = 3$ biological replicates). (D) qPCR analysis of SAT1 mRNA in PBS-stimulated and rmIL-25-stimulated ileum tissues as Fig. 2G ($n = 3$ biological replicates). (E) SAT1 knockdown efficiency was carried out by qPCR analysis in siNC-treated and siSAT1-treated HeLa cells for 24 h ($n = 3$ biological replicates). (F, G) $WT$_HeLa cells and $IL$-$25^{+/+}$ HeLa cells were transfected with siNC and siSAT1 for 24 h, and then they were infected with EV71 for 24 h. (F) qPCR was used to measure the VP1 mRNA level, and GAPDH was used as a housekeeping gene ($n = 3$ biological replicates). (G) $TCID_{50}$ was used to measure EV71 titers from the cell supernatant based on the result of F ($n = 3$ biological replicates). (H) qPCR analysis of SAT1 mRNA expression in lenti-shNC-Caco-2 and lenti-shSAT1#01/#02/#03/#04-Caco-2 cells, respectively ($n = 3$ biological replicates). (I, J) Lenti-shNC-Caco-2 and lenti-shSAT1#03-Caco-2 were both infected with EV71 for 24 h. qPCR and $TCID_{50}$ were used to detect the 5′UTR mRNA, and viral titers in EV71-infected Lenti-shNC-Caco-2 and lenti-shSAT1#03-Caco-2, respectively ($n = 3$ biological replicates). (K, L) HeLa cells were transfected with pVector and pSAT1 for 24 h, and then they were both infected with EV71 for 24 h. $TCID_{50}$, Western blotting and qPCR were used to detect the viral titers, VP1 protein and 5′UTR mRNA, respectively ($n = 3$ biological replicates). (M) $WT$_ HeLa cells and $IL$-$25^{+/+}$ HeLa cells were transfected with siNC and siSAT1 for 24 h, and then they were infected with EV71 for 24 h. Western blotting was used to detect the VP1, SAT1, and GAPDH. Quantification of western blotting bands based on the result of (M) showed the VP1 and SAT1 protein levels, respectively ($n = 3$ biological replicates). (N) $TCID_{50}$ was used to detect viral titers from the supernatant based on the result of (M). (O) Caco-2 cells were treated with rhIL-25 (10 ng/ml) alone or in combination with 10 μM DA for 2 h before EV71 infection. The supernatant was collected 24 h later, and viral titers were determined by TCID50 assay ($n = 3$ biological replicates). (P, Q) Lenti-shNC-Caco-2 cells and lenti-shSAT1#03-Caco-2 cells were infected with EV71/CVA16/CVB3 for 24 h, respectively. qPCR and $TCID_{50}$ were used to measure the 5′UTR mRNA and viral titers from lenti-shNC-Caco-2 cells and lenti-shSAT1#03-Caco-2 cells ($n = 3$ biological replicates). Data expression as Mean ± SD, the major statistical procedures applied were: Shapiro-Wilk test (B–Q), $F$-test (H, M, N–Q), Wilcoxon test (B–E), Student's $t$-test (F, G, I–L), Kruskal–Wallis (H), one-way ANOVA and Dunnett's $t$-test (M–Q). ns no significant difference; **$P < 0.01$ and ***$P < 0.001$. Source data are available online for this figure.

# Methods

### Reagents and tools table

| Reagent/Resource | Reference or source | Identifier or catalog number |
|---|---|---|
| **Experimental models** | | |
| AG6 mice | Prof. Qibin Leng | N/A |
| *B6.IL-25*$^{-/-}$ mice | Prof. Cheng Dong | Angkasekwinai et al, 2010 |
| **Recombinant DNA** | | |
| pCMV-SAT1-GFP | MIAOLING BIOLOGY | P37336 |
| pCMV-IL-25-GFP | SinoBiological Co., Ltd | Cat: HG10096-ACG |
| **Antibodies** | | |
| VP1 | Abcam, Inc | ab36367 |
| IL-25 | R&D | MAB1258-100 |
| DCLK1 | Abcam, Inc | ab31704 |
| SAT1 | Proteintech Group, Inc | Cat#: 10708-1-AP |
| TRPM5 | Proteintech Group, Inc | Cat#: 18207-1-AP |
| **Oligonucleotides and other sequence-based reagents** | | |
| QPCR primers | This study | Table 1 |
| Target sequences | This study | Table 2 |
| **Chemicals, enzymes, and other reagents** | | |
| Recombinant human IL-25 | Abclonal Biotech Co., Ltd | Cat#: RP01755 |
| Recombinant murine IL-25 | SinoBiological Co., Ltd | Cat#: 5013-M07H |
| **Software** | | |
| GraphPad Prism 9.0 | San Diego, CA, USA | https://www.graphpad.com/scientific-software/prism/ |
| **Other** | | |
| Intracellular Polyamine Detection Reagent | Funakoshi, Dongjing, Japan | Cat#: FDV-0020 |

## Reagents

Recombinant human IL-25 (rhIL-25) (Cat#: RP01755) was purchased from Abclonal Biotech Co., Ltd (Wuhan, China). Recombinant murine IL-25 (rmIL-25) (Cat#: 5013-M07H) was purchased from SinoBiological Co., Ltd (Beijing, China). The primary antibodies were diluted as follows: mouse anti-GAPDH (Cat#: 60004-1-1 g, 1:1000) was purchased from Proteintech Group, Inc (Wuhan, China); Rabbit-anti-DCLK1 (Cat#: ab31704, 1:100 for immunofluorescence staining) and Mouse anti-Enterovirus 71 antibody (ab36367, 1:1,000 for Western blotting and 1:100 for immunofluorescence staining) were both purchased from Abcam, Inc (Suzhou, China); Rabbit-anti-TRPM5 (Cat#: 18027-1-AP, 1:100) and SAT1 (Cat#: 10708-1-AP, 1:1000 for Western blotting and 1:100 for immunofluorescence staining) were purchased from Abclonal Biotech Co., Ltd (Wuhan, China). The antibody specific for IL-25 (MAB1258-100, 1:1000 for Western blotting, and 1:100 for immunofluorescence staining) was purchased from R&D.

## Cells

HeLa (human cervical cancer cell line), 293T (human embryonic kidney cell line), Caco-2 (human epithelial cell line), and HT-29 (human colorectal cell line) were obtained from ATCC and maintained in DMEM with 10% heat-inactivated FBS.

## Viral strains

EV71 BrCr and CVA16 were a gift from Prof. Bin Wu, Jiangsu Provincial Centers of Disease Control, China. Coxsackievirus B3 (CVB3) (Nancy strain) was kindly provided by Prof. Zhanqiu Yang, Wuhan University, China. According to previous studies (Lyu et al, 2024; Wu et al, 2023), EV71 was propagated in RD cells, and CVA16 and CVB3 were propagated in Vero cells. Viral titer was determined by $TCID_{50}$ assay. All experiments with EV71, CVA16,

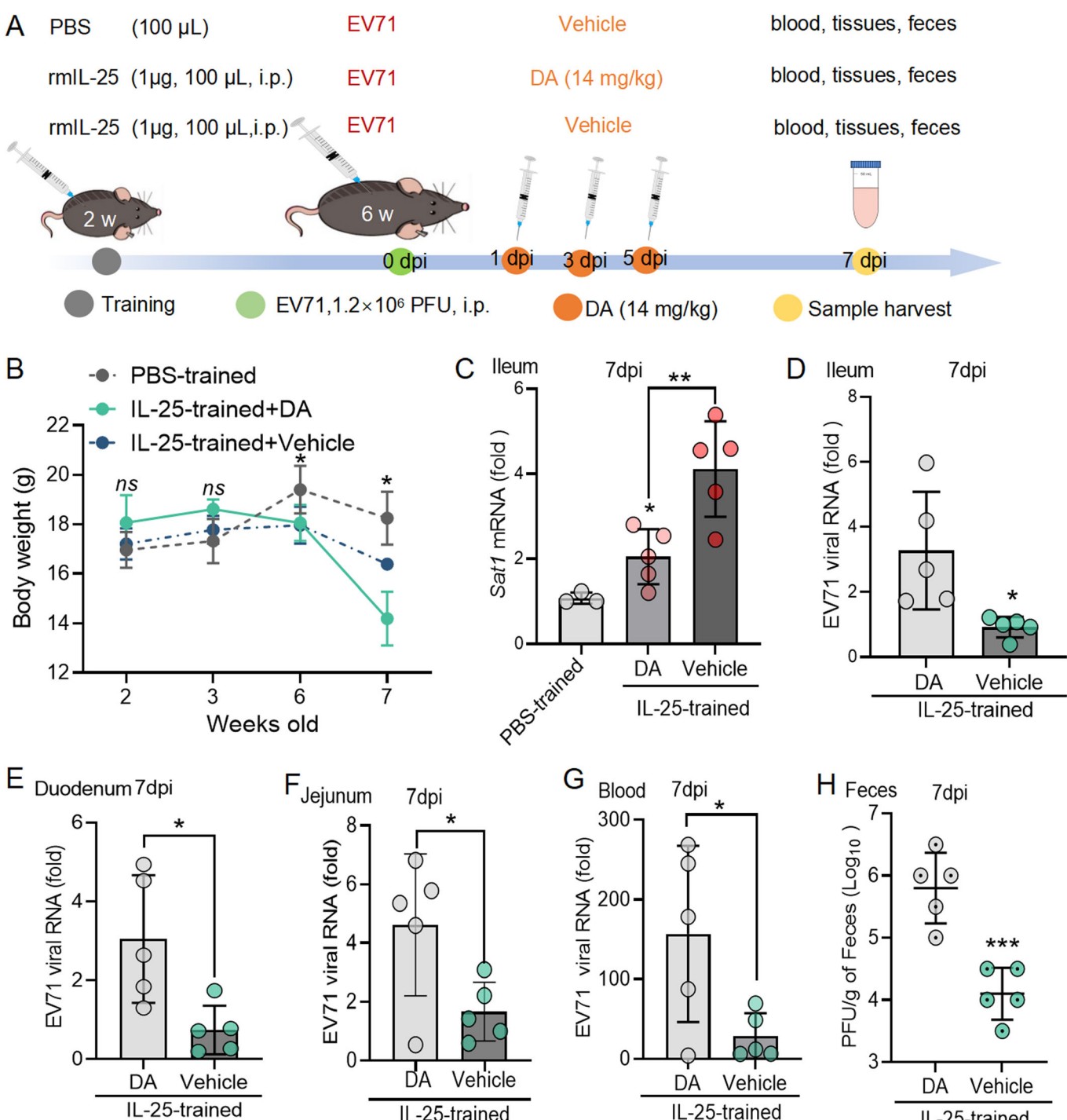

**Figure 7. DA abolished IL-25-mediated antiviral effects dependent on SAT1 expression.**

(A) The experiment design for (B–H) was shown. AG6 mice at 2 weeks old in infancy received rmIL-25 treatment (1 μg, three times, i.p.) for 1 week. When the rmIL-25-trained mice grew to 6 weeks old in the adult stage, they were infected with EV71 (1.2 × 10⁶ PFU, i.p.) on day 0, and then they were treated in the absence (*n* = 5) or the presence of DA (14 mg/kg, *n* = 5) on day 1, 3, 5 dpi. The mice were killed on 7 dpi. (B) The body weight of mice was counted from rmIL-25-trained mice in combination with DA or Vehicle. They were both compared with PBS-trained mice (*n* = 5), respectively. (C) qPCR analysis of *Sat1* mRNA expression in ileum tissues from IL-25-trained mice with DA or Vehicle in challenge of EV71 infection on 7 dpi. *$P < 0.05$, **$P < 0.01$, ***$P < 0.001$ vs. PBS-trained mice. (D–G) EV71 replication was monitored by qPCR analysis of viral RNA expression in the ileum, duodenum, jejunum tissues, and blood, respectively. (H) The PFU/mL of feces from IL-25-trained mice with DA or Vehicle in the challenge of EV71 infection on 7 dpi were measured by TCID$_{50}$ assay (*n* = 5). Data expression as Mean ± SD, the major statistical procedures applied were: Shapiro–Wilk test (B–H), *F*-test (B, C), repeated measures ANOVA (B), Kruskal–Wallis *H* (C), Student's *t*-test (D–H). ns no significant difference; *$P < 0.05$, **$P < 0.01$, ***$P < 0.001$. Source data are available online for this figure.

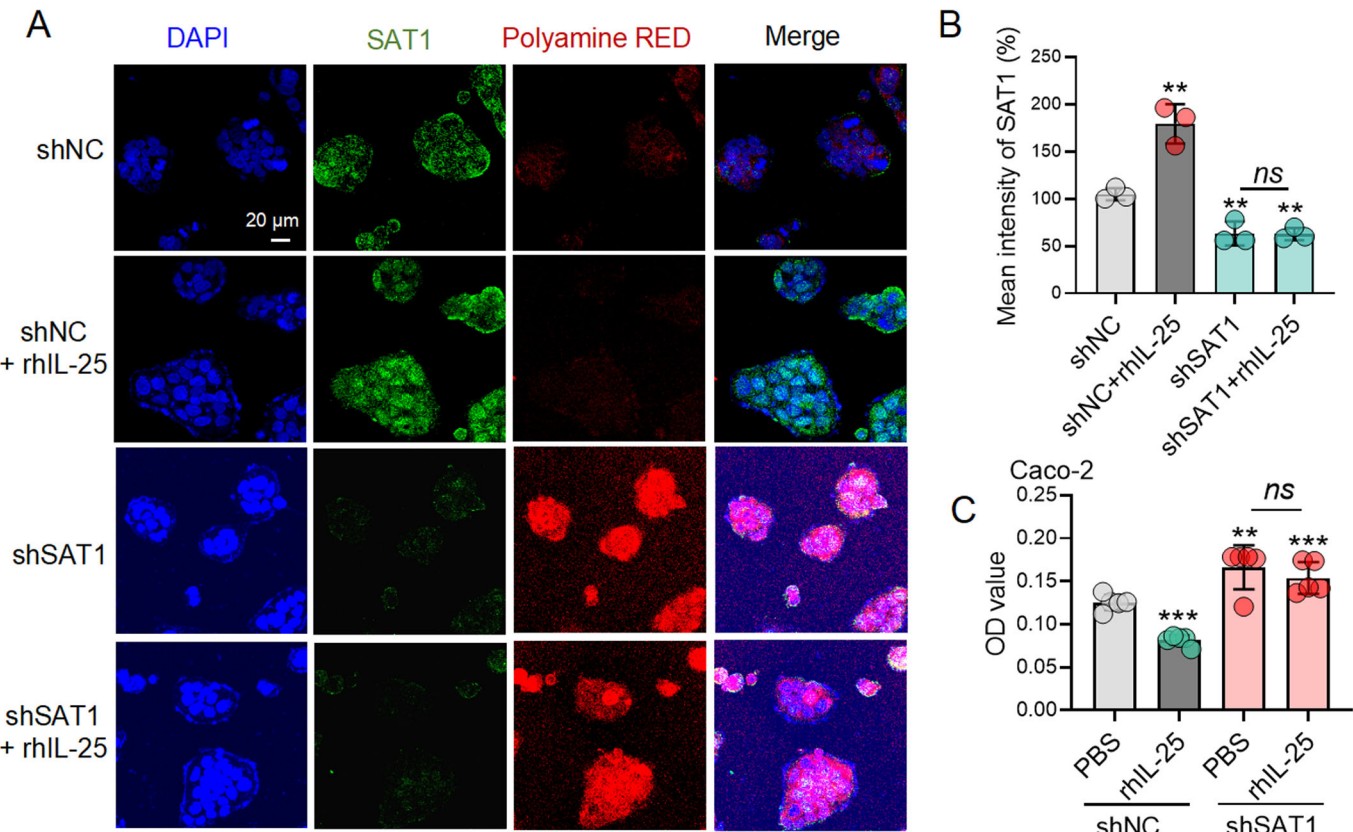

**Figure 8. IL-25-SAT1 axis negatively regulated the polyamine level dependent on SAT1 expression.**

(A) Lenti-shNC-Caco-2 cells and lenti-shSAT1#03-Caco-2 cells were treated with PBS or rhIL-25 (10 ng/mL) for 24 h, respectively. Cell imaging of Caco-2 cells that were incubated and treated with 30 μM of PolyamineRED for 10 min, stained by DAPI, and fixed. IF staining of SAT1 protein (green) and polyamine (red), respectively. Scale bar: 20 μm. (B) The fluorescence intensity of SAT1 was presented as mean ± SD ($n = 3$ biological replicates). (C) The relative fluorescence intensities normalized against the background were obtained by the Tecan Reader and presented as the OD values ($n = 5$ biological replicates). Data expression as Mean ± SD, the major statistical procedures applied were: Shapiro-Wilk test and $F$-test (B, C), one-way ANOVA and SNK-q test (B, C). ns no significant difference; $**P < 0.01$, $***P < 0.001$. Source data are available online for this figure.

and CVB3 were performed in the Medical School of Nanjing University.

## HeLa cells or Caco-2 cells stably expressing IL-25

To generate HeLa or Caco-2 cells that stably overexpressing IL-25, IL-25 lentivirus (overexpression) was produced according to our previous study from Fang Zhang (Zhang et al, 2022). The plasmid expressing IL-25 together with pGag/Pol/pRev/pVSVG were co-transfected into 293T cells using Lipo8000™ (Beyotime, C0533-1.5 mL). After 16 h, the medium was replaced with DMEM containing 2% FBS, 0.01 mmol/L cholesterol, 0.01 mmol/L L-α-phosphatidylcholine, Chemically Defined Lipid (1:1000 dilution), and 4 mmol/L L-glutamine. Supernatants were collected after 48 h. Lentiviral titers were determined according to a TCID$_{50}$ assay. The lentiviruses (MOI = 1) were used to infect HeLa cells or Caco-2 cells. After infection for 12 h, HeLa cells or Caco-2 cells were cultured with 5 mg/mL puromycin to produce the cells. An empty vector was used under the identical conditions as the control.

## Knockdown of IL-25 by lentivirus-mediated shIL-25#3 in Caco-2 cells

To generate Caco-2 cells that knockdown of IL-25, lentivirus-mediated shIL-25#3 was produced according to our previous study from Fang Zhang (Zhang et al, 2022). The plasmid knockdown IL-25 named shIL-25#3, together with pGag/Pol/pRev/pVSVG were co-transfected into 293T cells using Lipo8000™ (Beyotime, C0533-1.5 mL). After 16 h, the medium was replaced with DMEM containing 2% FBS, 0.01 mmol/L cholesterol, 0.01 mmol/L L-α-phosphatidylcholine, Chemically Defined Lipid (1:1000 dilution), and 4 mmol/L L-glutamine. Supernatants were collected after 48 h. Lentiviral titers were determined according to a TCID$_{50}$ assay. The lentiviruses (MOI = 1) were used to infect Caco-2 cells. After infection for 12 h, Caco-2 cells were cultured with 5 mg/mL puromycin to produce the cells. An empty vector named lentivirus-shNC was used under identical conditions as the control.

**Table 1.   Sequences of qPCR primers.**

| Gene name | Forward primer (5′ to 3′) | Reverse primer (5′ to 3′) | species |
|-----------|---------------------------|---------------------------|---------|
| IL-25 | CAGGTGTACAACCACTTGCC | TCCAGAAATGGGCAGAACTT | human |
| GAPDH | TGCACCACCAACTGCTTAGC | GGCATGGACTGTGGTCATGAG | human |
| SAT1 | GAAGAGGTGCTTCTGATCGTC | CTCACTCCTCTGTTGCCATTT | human |
| GAPDH | CCATCAACGACCCCTTCATTGACC | TGGTTCACACCCATCACAAACATG | mouse |
| IL-25 | ACAGGGACTTGAATCGGGTC | TGGTAAAGTGGGACGGAGTTG | mouse |
| DCLK1 | TACCGACGCTATCAAGCTGGAC | GGTAACGGAACTTCTCTGGTCC | mouse |
| TRPM5 | GGTGTTCACACTTCGGCTCATC | CCACAAGCCATACGCTCAGGAA | mouse |
| IL-9 | TCCACCGTCAAAATGCAGCTGC | CCGATGGAAAACAGGCAAGAGTC | mouse |
| 5′UTR | ATTGAGCTAGTTAGTAGTCCTCCG | AATAGCTCTGTTTGACACTGGATGG | N/A |
| EV71 | TAACTGCGGAGCACATACCC | ACGGACACCCAAAGTAGTCG | N/A |

**Table 2.   Target sequences.**

| Gene name | Sequences |
|-----------|-----------|
| shIL-25#01 | TCTTGGCAATGGTCATGGGAA |
| shIL-25#02 | ACAGTTCTCTCATTAGCCTTT |
| shIL-25#03 | TCCTGTAGGGCCAGTGAAGAT |
| shSAT1#01 | GCTTTGGCATAGGATCAGAAA |
| shSAT1#02/siSAT1 | CGACAAGGAGTACTTGCTAAA |
| shSAT1#03 | CCCGTGGATTGGCAAGTTATT |
| shSAT1#04 | TCTTCGTGATGAGTGATTATA |

## Knockdown of SAT1 by lentivirus-mediated shSAT1#03 in Caco-2 cells

To generate Caco-2 cells that knockdown of SAT1, lentivirus-mediated shSAT1#03 was produced according to our previous study from Fang Zhang (Zhang et al, 2022) The plasmid knockdown SAT1 named shSAT1#03, together with pGag/Pol/pRev/pVSVG were co-transfected into 293T cells using Lipo8000™ (Beyotime, C0533-1.5 mL). After 16 h, the medium was replaced with DMEM containing 2% FBS, 0.01 mmol/L cholesterol, 0.01 mmol/L L-α-phosphatidylcholine, Chemically defined lipid (1:1000 dilution), and 4 mmol/L L-glutamine. Supernatants were collected after 48 h. Lentiviral titers were determined according to a $TCID_{50}$ assay. The lentiviruses (MOI = 1) were used to infect Caco-2 cells. After infection for 12 h, Caco-2 cells were cultured with 5 mg/mL puromycin to produce the cells. An empty vector named shNC was used under the identical conditions as the control.

## Ethic statement

C57BL/6J male mice at 2 weeks old were purchased from Yangzhou Translational Medicine Animal Centre. $IL-25^{-/-}$ mice were kindly provided by Prof. Cheng Dong and Dr. Miao Xu and have been backcrossed to C57BL/6 background for more than six generations (Angkasekwinai et al, 2010; Xu et al, 2018). AG6 mice (IFN -α/β and γ receptor-deficient) with the C57BL/6 background were provided by Prof. Qibin Leng of Guangxi Medical University. They were bred and maintained in a pathogen-free environment and all animal operations were accordance with the protocols from Nanjing University Animal Care Committee and operated in accordance with the guidelines for Laboratory Animal Care and Use Committee.

## Animal model establishment

AG6 male mice at 2 weeks old were infected with EV71 BrCr strain ($1.2 \times 10^6$ PFU) and CVA16 ($2.0 \times 10^6$ PFU) by intraperitoneal injection, respectively. The uninfected mice were injected with an equal volume of DMEM containing 2% FBS. On days 3, 5, 7, and 14 after the EV71 infection, mice were anaesthetized, and 4% paraformaldehyde was used to fix the tissues. qPCR was used to analyze EV71 viral RNA levels in the duodenum, jejunum, and ileum tissues from the PBS group ($n = 5$) and EV71-infected group ($n = 5$) on day 7 post infection from the mice. H&E staining of ileum tissues from PBS group and EV71-infected group, respectively.

$IL-25$ knockout ($IL-25^{-/-}$) and wide type C57BL/6J male mice at 2 weeks old were both infected with EV71 BrCr strain by intraperitoneal injection ($1.2 \times 10^6$ PFU). For the control group (uninfected), mice were injected with an equal volume of the culture medium, which was used for the virus stock preparation (DMEM, 2% FBS). Seven days after the infection, mice were anesthetized and perfused with PBS, followed by 4% paraformaldehyde. The small intestine and colon tissues were removed immediately, fixed by 4% paraformaldehyde, and embedded in paraffin.

C57BL/6J male mice at 2 weeks old were given mouse IL-25 recombinant protein (rmIL-25) as previously described (von Moltke et al, 2016). Briefly, mice were injected with 0.4 μg ($n = 6$) or 1.0 μg ($n = 6$) rmIL-25 on days 1, 3, and 5 by intraperitoneal injection, respectively. The mice treated with an equal volume of PBS ($n = 6$) were considered as a control. The ileum tissues were harvested for staining on day 7.

AG6 male mice ($n = 12$) at 2 weeks old were treated with rmIL-25 protein on days 1, 3 and 5 (1.0 μg, three times; i.p.) for 1 week by intraperitoneal injection. AG6 mice were injected with an equal volume of PBS ($n = 12$). Before they were infected with EV71, ileum tissues were collected at 3 weeks old ($n = 6$) and 6 weeks old ($n = 6$), respectively. When mice grew to adulthood, they were both intraperitoneally infected with EV71 ($1.2 \times 10^6$ PFU) ($n = 6$) for 1 week. The blood of EV71-infected adult mice was collected and

analyzed by qPCR. The small intestine and colon tissues were collected and fixed by 4% paraformaldehyde.

AG6 male mice ($n = 6$) at 2 weeks old were infected with EV71 infection ($1.2 \times 10^6$ PFU; i.p.) on day 0. AG6 male mice ($n = 6$) at 2 weeks old were treated with rmIL-25 protein (1.0 μg, three times, i.p.) for 1 week as a positive control. AG6 male mice at 2 weeks old were injected with an equal volume of PBS ($n = 6$). When the three group mice grew to adulthood (at 6 weeks old), they were both infected with EV71 ($1.2 \times 10^6$ PFU; i.p.) on day 0. The blood of EV71-infected adult mice was collected and analyzed by qPCR on day 7 dpi. The small intestine and colon tissues on day 7 dpi were collected and fixed by 4% paraformaldehyde.

AG6 male mice at 2 weeks old were treated with rmIL-25 protein (1.0 μg, three times; i.p.) on days 1,3, 5. For the control group, AG6 male mice were injected with an equal volume of PBS. When mice grew to adulthood at 6 weeks old, IL-25-trained mice were infected with EV71 ($1.2 \times 10^6$ PFU) for 1 week in the presence or in the absence of DA (14 mg/kg. three times). And PBS-trained mice were also infected with EV71 ($1.2 \times 10^6$ PFU) for one week. The blood and small intestinal tissues were collected and fixed by 4% paraformaldehyde. The feces were also collected and measured by a $TCID_{50}$ assay.

## Western blotting

Proteins were separated by 12% SDS-PAGE and transferred to PVDF membranes. After blocking the membranes with 2% BSA for 30 min and incubating overnight with primary antibodies, goat-anti-mouse IgG or donkey-anti-rabbit IgG was added to the membranes after three washes with PBST and incubated for 1 h. The expression levels of the target proteins were quantified by LI-COR Odyssey scanning. The density of each band was first normalized to the loading control and then to the control.

## In-cell Western blotting

For the virus yield assay, in-cell Western blotting was performed to measure the viral titers from the cell supernatant on fresh Vero cells. The viral titers were determined by scoring the plaque-forming units (PFU) with modifications as previously described (You et al, 2023). The diluted virion-containing medium was dispensed on fresh Vero cells. The cells were fixed with 4% paraformaldehyde and permeabilized in 0.1% Triton-X100. Cell monolayers were blocked for 60 min and then incubated with primary antibodies for 1 h. After cell monolayers were washed 5 times with washing buffer and stained in the secondary antibody for 1 h and scanned in Odyssey Infrared Imager. DRAQ5 staining was the normalized control.

## Real-time PCR (qPCR) and RNA-seq

Total RNA was first isolated with trizol reagent and reverse transcribed into cDNA using a reverse transcription kit. Real-time PCR was performed with the SYBR Green Supermix and primers, as shown in Table 1. For RNA analysis, HeLa cells ($WT\_$HeLa cells) and HeLa cells stably expressing IL-25 ($IL-25^{+/+}\_$HeLa cells) were uninfected and infected with EV71 at an MOI of 0.2 for 24 h. Total RNAs were isolated and RNA-seq library was prepared by LC-Bio Technologies (Hangzhou) Co., Ltd. Viral gene copy number was

counted by qPCR analysis. The primers set for EV71 viral RNA and other specific primers were also listed in Table 1. The standard curve of the plasmid pT7-EV71-5'UTR was generated as a control. siRNA or shRNA sequences were described in Table 2.

## Histology, immunohistochemistry (IHC), and immunofluorescence staining

H&E staining was performed with some modifications as previously described. For immunofluorescence, the mouse intestine tissue was fixed, blocked, and then stained with antibodies specific for IL-25 (MAB1258-100, R&D), VP1 (ab36367, Abcam), DCLK1 (ab31704, Abcam), and TRPM5 (Cat#:18027-1-AP). Fluorescence images were taken with a ZEISS microscope, and the number of the tuft cells were analyzed by imageJ analysis software.

## Tuft cell quantification

In uninfected and EV71-infected mice, a 10 cm section of small intestines, including duodenum, jejunum, and ileum was harvested, and IHC was performed to quantify the number of tuft cells. The small intestine about 10 cm was harvested and stained for DCLK1 and TRPM5 markers as previously described (von Moltke et al, 2016). A $4 \times 4$ grid of images was collected (×200). The area of $DCLK1^+/TRPM5^+$ and DAPI staining were calculated using ImageJ. The percent of tuft cells was calculated as $DCLK1^+/TRPM5^+$ positive cells/DAPI staining cells. The graph showed the distribution of $DCLK1^+$ tuft cells in ileum tissues in naive and EV71-infected mice on days 3, 5, 7, and 14 post infection. Tuft cells were counted in the crypt and villus compartments of $n = 50$ crypt–villus units per mouse with three to five mice per condition. Means of villus/crypt ratio of tuft cell numbers are shown.

## Intracellular polyamine imaging and detecting reagent

The polyamine red assay was performed by using the total Polyamine Assay kit (PolyamineRED, Intracellular Polyamine Detection Reagent, Cat#: FDV-0020). PolyamineRED possesses the cell membrane permeability and reacts with polyamines inside the cell and couples TAMRA to polyamines. Since TAMRA loses its membrane permeability when it binds to polyamines, therefore, the unreacted PolyamineRED can be removed by washing with PBS. Total polyamines can be semi-quantified by intracellular fluorescence intensity by the reader. The distribution of intracellular polyamines can be observed by a ZEISS microscope. The relative fluorescence intensities were analyzed by imageJ analysis software. This TAMRA (tetramethylrhodamine)-conjugated derivative of glycine propagyl ester specifically reacts with linear primary alkylamine and has cell-penetrating properties, specifically reacts with polyamines inside the cells and labeled polyamines with red fluorescent dye TAMRA. shNC-Caco-2 cells and shSAT1#03-Caco-2 cells were treated with rhIL-25 (10 ng/mL) or PBS for 24 h, After then, they were treated with 30 μM of PolyamineRED for 10 min. After incubation, cells were washed three times by PBS, followed by formalin fixation. The fixed Caco-2 cells were further incubated with SAT1 antibody and DAPI staining. Lastly, they were incubated with Goat-anti-Rabbit-488-FITC. The TAMRA fluorescence signal was determined at 546 nm. Fluorescence images were taken with a ZEISS microscope.

**The paper explained**

**Problem**
Intestinal tuft cells, a type of innate immune cell rarely found in the intestines, play a crucial role in the host's defense against intestinal pathogens. While innate immune cells have been recognized as trainable in recent years, there is still uncertainty regarding the training ability of tuft cells.

**Results**
In this study, we initially established an enteroviral-infected mouse model and found that intestinal tuft cells were significantly increased. Furthermore, EV71 infection-mediated IL-25 production contributed to drive tuft cell expansion as well as IL-25-trained intestinal tuft cells obtained the immune memory in mice. Upon re-challenge, IL-25-trained tuft cells were quickly increased and displayed a stronger IL-25-SAT1 axis response and exhibited a more powerful antiviral effect as compared to naïve tuft cells. Furthermore, IL-25 supported the host defense via upregulating SAT1 expression, together with a lower polyamine level, which affected a broad enteroviral replication.

**Impact**
Our current study provides a novel perspective on intestinal tuft cell training during enteroviral infections. This observation may pave the way for exploring a therapeutic strategy to combat enteroviral replication by manipulating IL-25 production to train tuft cells in infancy.

## Statistical analysis

In the present study, we all employed a random number table to perform randomization. All animal experiments were performed and analyzed in a blinded manner. Treatment groups were assigned in a randomized fashion. Each mouse was assigned a temporary random number within the weight range and they were given their permanent numerical designation in the cages when they were randomly divided into each group. For each group, a cage was selected randomly from the pool of all cages. All data were collected and analyzed by two observers who were not aware of the group assignment or treatment of the mice. Statistical analyses were performed using GraphPad Prism 9.0 software (GraphPad, San Diego, CA, USA). All data were initially determined to be normally distributed or not by *the* Shapiro-Wilk test. For data that conforms to a normal distribution, comparisons between two groups were analyzed using a two-tailed Student's *t*-test or paired *t*-test. One-way ANOVA followed by Dunnett's *t*-test was used to compare more than two data groups, which were run only if *F*-test achieved $P > 0.05$, and there was no significant variance in homogeneity. Nonparametric tests were used for datasets that did not conform to a normal distribution or with *F*-test $P < 0.05$. The Kruskal–Wallis *H*-test was used to compare more than two data groups. Wilcoxon test was used to compare the repeated measures data of the two groups or the two independent samples. *P* values were provided as *$P < 0.05$, **$P < 0.01$, ***$P < 0.001$. The exact *p* values were provided in Appendix Table S1.

## Data availability

The datasets produced in this study are available in online and open-access databases. RNA sequencing data: GSE217085 (https://www.ncbi.nlm.nih.gov/geo/query/acc.cgi?acc=GSE217085). The

source data of this paper are collected in the following database record: biostudies:S-SCDT-10_1038-S44321-024-00128-9.

## Peer review information

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

## Acknowledgements

We thank Professor Cheng Dong (School of Medicine, Tsinghua University, Beijing, China) for providing *IL-25$^{-/-}$* mice. AG6 mice were a gift from Professor Qibin Leng (Affiliated Cancer Hospital & Institute of Guangzhou Medical University). EV71 BrCr strain and CVA16 were provided by Prof. Bin Wu, Jiangsu Provincial Centers of Disease Control. Coxsackievirus B3 (CVB3) (Nancy strain) was kindly provided by Prof. Zhanqiu Yang (Wuhan University, China). This work was supported by the National Natural Science Foundation of China (81900823 to CDY and 31970149 to WZW).

## Author contributions

**Deyan Chen**: Conceptualization; Data curation; Formal analysis; Investigation; Methodology; Writing—original draft. **Jing Wu**: Conceptualization; Data curation; Software; Methodology; Writing—original draft. **Fang Zhang**: Data curation; Methodology; Writing—original draft. **Ruining Lyu**: Data curation; Methodology. **Qiao You**: Methodology. **Yajie Qian**: Data curation; Methodology. **Yurong Cai**: Data curation; Methodology. **Xiaoyan Tian**: Data curation; Methodology. **Hongji Tao**: Data curation; Methodology. **Yating He**: Data curation; Methodology. **Waqas Nawaz**: Writing—review and editing. **Zhiwei Wu**: Conceptualization; Supervision; Funding acquisition; Project administration; Writing—review and editing.

Source data underlying figure panels in this paper may have individual authorship assigned. Where available, figure panel/source data authorship is listed in the following database record: biostudies:S-SCDT-10_1038-S44321-024-00128-9.

## Disclosure and competing interests statement

The authors declare no competing interests.

# Expanded View Figures

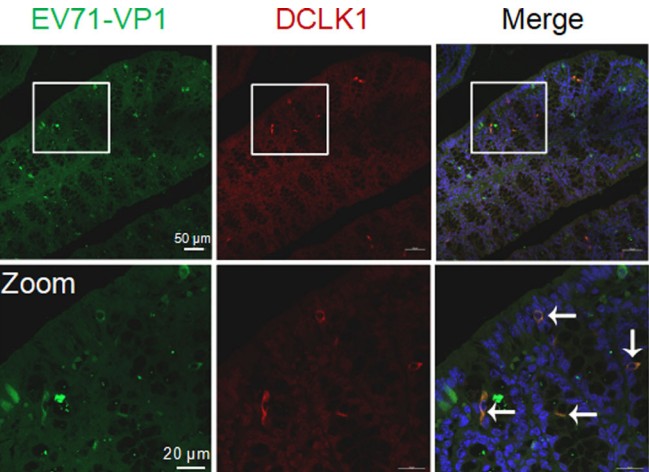

**Figure EV1.  EV71 infected tuft cells.**

AG6 male mice were infected with EV71 for 7 dpi, and ileum tissues were collected. Immunofluorescence assays were used to detect EV71-viral protein VP1 expression (Green) and DCLK1 expression (Red). Scale bar: 50 μm. White arrows indicated the co-expression of VP1 and DCLK1. Areas in white squares were shown in threefold magnification on the bottom row. Scale bar: 20 μm. Source data are available online for this figure.

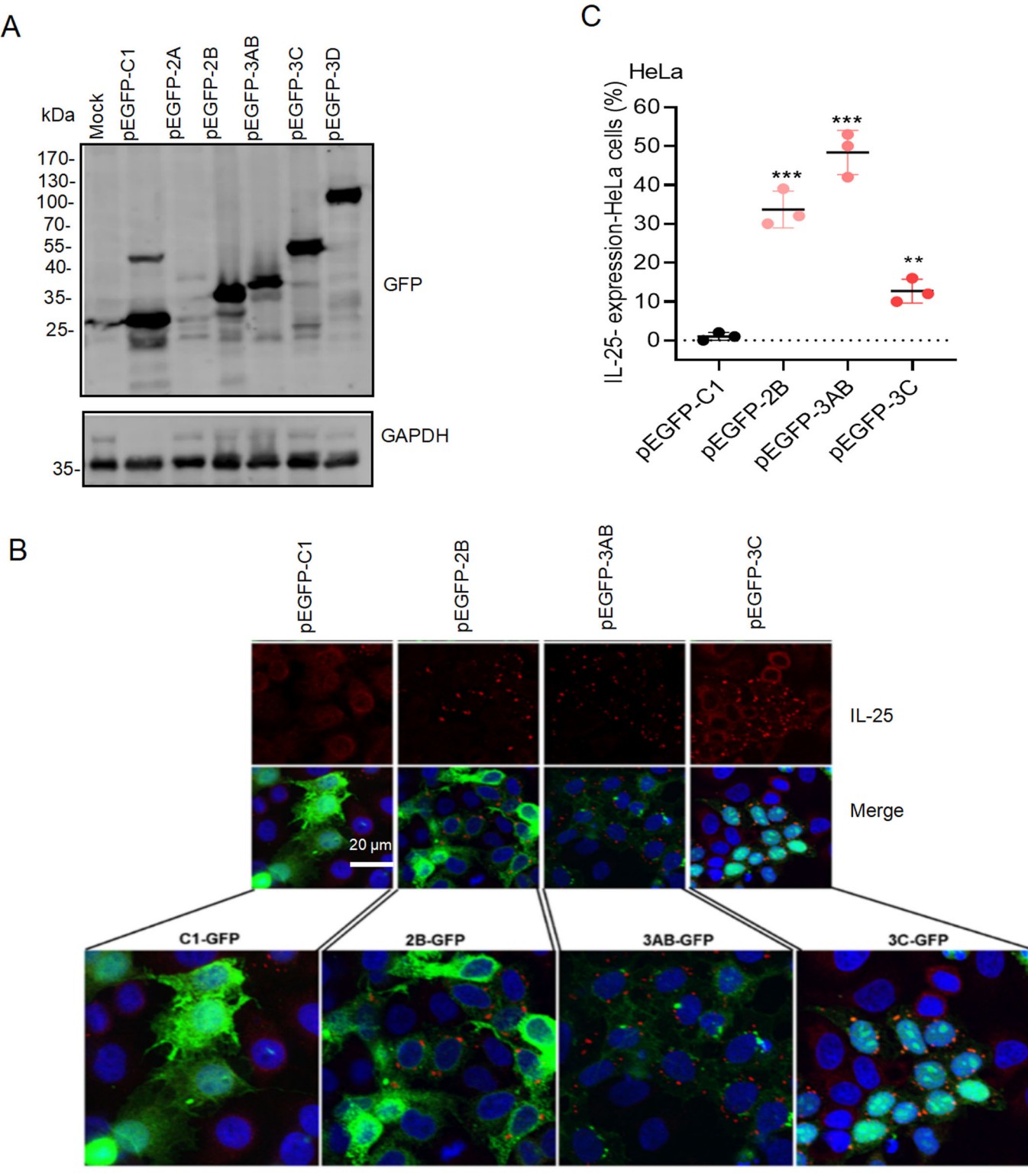

**Figure EV2. EV71 viral protein induced IL-25 production.**

(A) Western blotting detected GFP and GAPDH expression in the presence of EV71 2A, 2B, 3AB, 3C, and 3D in 293T cells for 24 h. (B) Immunofluorescence staining of IL-25-expressing HeLa cells in the presence of EV71 2B, 3AB, and 3C, respectively. Scale bar: 20 μm. (C) quantitative analyses of the number of IL-25-secreted-HeLa cells from (B) were performed. Data were presented as mean ± SD ($n = 3$ biological replicates). The major statistical procedures applied were: Shapiro-Wilk test and $F$-test (B, C), one-way ANOVA and SNK-q test (B, C). **$P < 0.01$, ***$P < 0.001$ vs. the control cells (p-EGFP-C1-group). Source data are available online for this figure.

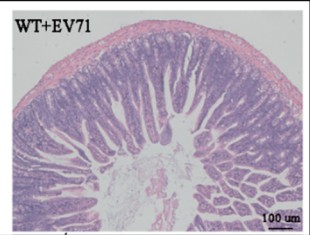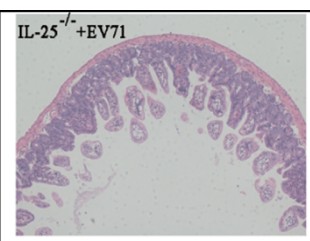

**Figure EV3.   Ileum tissues from EV71-infected IL-25$^{-/-}$ mice and WT mice at 7 dpi were stained with H&E.**

Scale bar: 100 μm. Source data are available online for this figure.

