## [Peer Review File · EMBO Molecular Medicine]

Trained Immunity of Intestinal Tuft Cells Enhances Host Defense against Enteroviral Infections

Deyan Chen, Jing Wu, Fang Zhang, Ruining Lyu, Qiao You, Yajie Qian, Yurong Cai, Xiaoyan Tian, Hongji Tao, Yating He, Waqas Nawaz, and Zhiwei Wu

Corresponding author: Zhiwei Wu (wzhw@nju.edu.cn)

Review Timeline:

Submission Date:	18th Nov 23
Editorial Decision:	10th Jan 24
Revision Received:	4th Jul 24
Editorial Decision:	15th Jul 24
Revision Received:	23rd Jul 24
Accepted:	12th Aug 24

Editor: Zeljko Durdevic

Transaction Report:

10th Jan 2024

Dear Dr. Chen,

Thank you for the submission of your manuscript to EMBO Molecular Medicine. We have now received feedback from the three reviewers who agreed to evaluate your manuscript. All three referees recognize potential interest of the study but also raise important and partially overlapping criticism that should be addressed in a major revision. Considering that the revision will require extensive experimentations we think six months rather than three months would be more appropriate to provide the complete revision. If you would like to discuss further the points raised by the referees, I am available to do so via email or video. Let me know if you are interested in this option.

We would welcome the submission of a revised version within six months for further consideration. Please let us know if you require longer to complete the revision.

I look forward to receiving your revised manuscript.

Yours sincerely,

Zeljko Durdevic

We require:

- 1) A .docx formatted version of the manuscript text (including legends for main figures, EV figures and tables). Please make sure that the changes are highlighted to be clearly visible.
- 2) Individual production quality figure files as .eps, .tif, .jpg (one file per figure). For guidance, download the 'Figure Guide PDF': (<https://www.embopress.org/page/journal/17574684/authorguide#figureformat>).
- 3) A .docx formatted letter INCLUDING the reviewers' reports and your detailed point-by-point responses to their comments. As part of the EMBO Press transparent editorial process, the point-by-point response is part of the Review Process File (RPF), which will be published alongside your paper.
- 4) A complete author checklist, which you can download from our author guidelines (<https://www.embopress.org/page/journal/17574684/authorguide#submissionofrevisions>). Please insert information in the checklist that is also reflected in the manuscript. The completed author checklist will also be part of the RPF.
- 5) Please note that all corresponding authors are required to supply an ORCID ID for their name upon submission of a revised manuscript.

6) It is mandatory to include a 'Data Availability' section after the Materials and Methods. Before submitting your revision, primary datasets produced in this study need to be deposited in an appropriate public database, and the accession numbers and database listed under 'Data Availability'. Please remember to provide a reviewer password if the datasets are not yet public (see <https://www.embopress.org/page/journal/17574684/authorguide#dataavailability>).

13) Author contributions: You will be asked to provide CRediT (Contributor Role Taxonomy) terms in the submission system. These replace a narrative author contribution section in the manuscript.

14) A Conflict of Interest statement should be provided in the main text.

Please also suggest a striking image or visual abstract to illustrate your article as a PNG file 550 px wide x 300-800 px high.

**** Reviewer's comments ****

Referee #1 (Comments on Novelty/Model System for Author):

The manuscript focuses on a relevant clinical and biological problem. While the data seem clear, statistical aspects make unclear the robustness of the conclusions, as detailed in my suggestions to the author.

Referee #1 (Remarks for Author):

This is an interesting study that aims to study whether tuft cells in the intestine can undergo long-term functional reprogramming similar to trained immunity induced in other innate immune cells. The authors show that retroviruses can induce such long-term effects, with trained tuft cells being able to produce significantly more IL-25 with anti-viral activities compared to untrained cells. Interestingly, IL-25 itself can train tuft cells to produce more IL-25, a positive loop that is suggested to be involved in the resistance to the virus.

Comments:

1. My main concern lies with the small number of animals in many of the experiments shown in various figures. Often the number of mice were $n=3$, which raises the question of robustness of the results. I think that the number of mice per experiment should be increased to validate the findings.
2. Similarly, the main results should be validated in independent experiments. Reproducibility is an important aspect that should be demonstrated.
3. Student t-test was used for comparisons, while it is likely that many data do not have a normal distribution (cytokines are known not to be normally distributed). Non-parametric tests should be done.
4. While the trained immunity effect on tuft cells seems to be robust, an important question is whether this effect depends on other cell types in the intestine. Can the authors speculate if either local macrophages or lymphocytes may impact this training effect on tuft cells?

Referee #2 (Comments on Novelty/Model System for Author):

In this manuscript entitled "Trained Immunity of Intestinal Tuft Cells during Infancy Enhances Host Defense against Enteroviral Infections in Mice", Chen et al. describe that intestinal tuft cells expanded during EV71 infection. This expansion was associated with increased levels of IL-25. Likewise, recombinant IL-25 stimulation also led to tuft cell expansion. Tuft cell numbers decreased back to baseline after cessation of the infection, however, upon a secondary infection, they expanded strongly and quickly, and enhanced antiviral gene expression.

The authors used both mouse in vivo and cell line models to functionally and mechanistically address the main questions of why infants are highly susceptible to EV71, the role of tuft cells in immunity to EV71, and whether tuft cells can acquire innate memory.

Both the functional and mechanistic studies are carried out well, however the functional part of the study lacks key experiments. Moreover, in the current version of the manuscript, the authors do not provide an explanation for the maintenance and importance of tuft cells in memory immunity against EV71.

Major comments:

1. The authors aim to address the important question why infants are highly susceptible to EV71 infection, while older children and adults are not. Their hypothesis is that the tuft cells of older children and adults are trained through infection at infant age and therefore provide increased and faster antiviral immunity upon re-infection. This hypothesis is interesting. However, the

study lacks basic experiments that provide the rationale for this research. Such as: EV71 infection in infant vs. adult wildtype mice. EV71 infection in adult mice that have or have not been infected/rIL-25-treated during infancy. It is not clear, why the initial experiment has been conducted in Typ I/II IFN KO-mice, not WT mice at the age of 2 weeks.

2. In Fig 2 and 4, the authors elegantly describe how EV71 infection, IL-25 secretion, transient expansion of tuft cells, and increased protection against EV71 infection are linked. The ultimate experiment to prove this link would however be the infection of IL-25^{-/-} mice with EV71 and subsequent reinfection, measurement of disease score and survival. Studies in (immortalized) cell lines are somehow confusing in this context. In fact, they open up more questions on the source of IL-25 and antiviral molecules. What would be the relevance of these cell types in the intestines, particularly in the ileum? Focusing on in vivo mouse studies by IF or FACS will certainly be helpful to carefully delineate the real importance of tuft cells (and their potential memory function) in protection against EV71 infection. Moreover, the authors need to investigate whether increased cytokine and/or antiviral molecule levels (or their gene expression) is simply due to the expansion of the producing cells, or due to enhanced production capabilities of each single cell. This should be investigated through staining in isolated tuft cells from mice as well as upon in vitro re-stimulation.

3. Tuft cells are rare cells in the intestines. The authors should elaborate on this in the introduction, such as their frequency in the different parts of the intestines as well as their development and function. Specifically, as the half-life of most tuft cells is only 4-5 days, how do the authors propose that training is maintained for >4 weeks in mice? Is it through training in their progenitor cells or are long-lived tuft cells reprogrammed that specifically drive the secondary response? These questions need to be addressed using tuft cell-specific knockout or reporter mice (see e.g., Chandrasekaran et al., *Oncotarget*, 2015).

Minor Comments:

1. Please proofread manuscript by a native English speaker. While the English is ok throughout most of the text, the choice of especially verbs is uncommon and confusing in multiple places.
2. Introduction, ll. 113: More cell types have been shown to acquire immune training. Furthermore, differentiate between listing the signaling pathway (cytokines) vs the inducer (e.g., pathogen) of immune training.
3. Why has an intracranial infection of 2-week old AG6 mice been used for EV71 and CVA16 or is this a misspelling in the Material and Methods section?
4. What route of injection was used for rIL-25 in C57BL/6 mice?
5. In models, please indicate age and sex of the mice.
6. Fig 3 has been mislabelled in the text as Fig 4.
7. Fig 1G+H, what is the timepoint of this cytokine measurement?
8. Fig 1A and I, it is not clear from the models if the infection/recombinant cytokine injections were only once (on d0) or on multiple days?
9. Fig 1 J, K, L, M do not show different timepoints as indicated in Fig 1I. Please note, what time point is shown for these analyses.
10. Fig 2 M-T: It is not clear, what the "fold change" of mRNA expression relates to. In general, it is more informative to provide the total values instead of the fold changes.

Referee #3 (Comments on Novelty/Model System for Author):

Despite the chosen grades, there are major limitations to this manuscript as indicates below.

1. The manuscript text and structure as well as figure illustration is hard to follow, the language should be revised.
2. Important experimental information on virus strains and tools is lacking

Referee #3 (Remarks for Author):

Chen et al. use a type I and II receptor deficient mouse model to study the effect of IL-25 on enteric enterovirus 71 (EV71) replication. They show that (i) IL-25 is sufficient to increase Tuft cells and required for EV71 induced Tuft cell increase at 2 weeks post infection; (ii) rIL-25 treatment or administration of a non-lethal EV71 dose leads on a transient Tuft cell number increase (peak at 2 week, baseline at 6 weeks). Infection 6 weeks after infant (2 week-old mice) rIL-25 treatment or EV71 infection leads to enhanced IL-25 levels and Tuft cell numbers (over non IL-25/EV71 infected controls) and a reduced EV organ load; (iii) viral protein expression in epithelial cell lines induces IL25 expression and secretion; (iv) IL-25 overexpression/rIL25 administration/transgene expression of IL-25 using a CMV system suppresses viral protein (VP1) expression and EV71 infection in IL-25 KO mice leads to higher virus titers and no tuft cell expansion; IL-25 transfected HeLa cells upregulate SAT1 (among other genes) and IL-25 treatment of ileal tissue increases SAT1 expression 2-fold, knock-down of SAT1 leads to increased EV71 replication in Caco2 cells, SAT1 overexpression leads to decreased viral gene expression in HeLa cells. SAT1 knockdown/SAT1 inhibitor abrogate the protective IL-25 effect on viral replication; (v) SAT1 overexpression leads to polyamine depletion (known, polyamines required for RNA virus replication) and (vi) diminazene aceturate (DA, a SAT1 inhibitor) administration abolishes the trained effect of early life rIL-25 treatment in vivo.

Generally, the data shown are timely, medically relevant and interesting. However, there are a number of issues that make this manuscript not suitable for publication in the present form.

Major points:

1. Please define the target cell for EV71 in the AG6 model, are tuft cells infected?
2. Please include the statistical test used for each panel (legend text). If parametric tests are used test for normal distribution.
3. Please discuss the source of IL-25 under in vivo conditions.
4. Please discuss the AG6 mouse model that lack IFN type I and II receptor expression (in the material and methods section called alpha, beta and gamma receptor, give a more detailed description). How could the lack of IFN I and II interfere with IL-25 and tuft cell numbers? Also type III IFN is still present in these mice and should in the epithelium confer resistance, discuss why the model is suitable. Notably SAT1 was reported to be IFN-induced (Mounce et al., Cell Host Microbe, 2016).
5. The definition of the time window (< and > 2 weeks) that facilitates the described trained immunity phenotype would add to the manuscript.
6. Improvement of the text by a native English-speaking person would significantly improve the clarity of the manuscript.
7. Does IL-25 overexpression in intestinal epithelial cell lines leads to cytokine secretion in the supernatant? Do epithelial cells in vivo (cell lines in this study) express the IL-25 receptor?
8. Fig. 3D is unclear (in legend panel D is described to show an IF) how can IL-25 and VP1 be shown on one single membrane?
9. What was the rationale to select SAT1 for further studies?
10. IL25 blockade was shown to augment antiviral immunity against (RNA) respiratory viruses (Williams et al., 2022, Commun Biol), Please discuss.
11. Please better explain the polyamine red assay. Is polyamine red secreted to allow OD measurements (not explained in M&M section).
12. The quantification of Tuft cells is unclear? L521: How many sections, fields of view etc. were analyzed?
13. Please better describe the virus strains used. The manuscript indicates the EV71 BrCr and CVA16 strains but which strain has been used for which panel is not indicated. The reference Chen et al. indicated in the manuscript does not give any further information. The reference Fu et al. refers to a EV71 Fuyang0805 strain that is not used in this study. Please clarify.

Minor points:

1. Please discuss the effect of cGas which has been shown to be controlled by polyamine metabolism (Zhao et al., Immunity, 2023)
2. Figure 1 C, please quantify villus lengths.
3. Please add bars in all figure panels.
4. Please comment on the role of ILC2s which respond to IL-25 and may be involved in the induction of tuft cell hyperplasia
5. Paragraph 2.3 refers to Fig. 3 but indicates Fig. 4.
6. Fig. 1 I-O may be moved to the supplemental information.
7. Fig. 4 N-T may be moved to the supplemental information as well.
8. Fig. 5 V-W, indicate what color indicates what.
9. Fig. 7 B, please provide body weight not as difference but as weight (g) per animals.
10. The color used could be harmonized to improve clarity.

point by point response letter

***** Reviewer's comments *****

Response: Thank you very much for your positive response and the valuable feedback provided by the reviewers for our manuscript. We greatly appreciate their recognition of the study's potential significance and have carefully considered their critical comments. We are fully committed to address each comment raised by the reviewers in a systematic and comprehensive manner, as detailed below.

Referee #1 (Comments on Novelty/Model System for Author):

The manuscript focuses on a relevant clinical and biological problem. While the data seem clear, statistical aspects make unclear the robustness of the conclusions, as detailed in my suggestions to the author.

Response: Thanks for your valuable comments. We have re-analyzed all the data used in this manuscript and sought the help of statistical professionals to standardize the use of statistical methods.

Referee #1 (Remarks for Author):

This is an interesting study that aims to study whether tuft cells in the intestine can undergo long-term functional reprogramming similar to trained immunity induced in other innate immune cells. The authors show that retroviruses can induce such long-term effects, with trained tuft cells being able to produce significantly more IL-25 with anti-viral activities compared to untrained cells. Interestingly, IL-25 itself can train tuft cells to produce more IL-25, a positive loop that is suggested to be involved in the resistance to the virus.

Response: We sincerely appreciate the positive feedback from the reviewer #1 on our manuscript. The reviewer pointed out several issues, which are very helpful for us to improve the manuscript. According to these valuable advice, we have amended the relevant parts of the manuscript and addressed the questions raised by the reviewer #1, which was shown as below:

Comments:

1. My main concern lies with the small number of animals in many of the experiments shown in various figures. Often the number of mice were $n=3$, which raises the question of robustness of the results. I think that the number of mice per experiment should be increased to validate the findings.

Response: The reviewer's concern is well taken. We increased the number of mice to 5-6, and performed the experiments again to confirm the conclusion in the revised manuscript according to your suggestions as shown in Figure 2H-K (originally presented as Figure 1 I-M), Figure 3A-E

(originally presented as Figure 2A-E) and Figure 5A-C (originally presented as Figure 4O-P) in the revised manuscript.

2. Similarly, the main results should be validated in independent experiments. Reproducibility is an important aspect that should be demonstrated.

Response: We sincerely thank your careful reading and agreed with your points. In this revision, we rechecked each experiment and confirmed the independent experiments in revised manuscript.

3. Student t-test was used for comparisons, while it is likely that many data do not have a normal distribution (cytokines are known not to be normally distributed). Non-parametric tests should be done.

Response: Thanks for your professional comments. We have re-analyzed all the data used in this study, and sought the help of statistical professionals to standardize the use of statistical methods. In this revision, we mentioned the primary statistical methods used for each set of data in the figure legend. And we revised the description of statistical analysis in METHODS section [Lines:654-664], which was shown as below:

Statistical analyses were performed using GraphPad Prism 9.0 software (GraphPad, San Diego, CA, USA). All data were initially determined to be normally distributed or not by the *Shapiro-Wilk* test. For data that conforms to a normal distribution, comparisons between two groups were analyzed using two-tailed *Student's t-test*. *One-way ANOVA* followed by *Dunnett'-t test* was used to compare more than two data groups, which were run only if *F-test* achieved $P > 0.05$, and there was no significant variance in homogeneity. Nonparametric tests were used for data sets that did not conform to a normal distribution. *P* values were provided as * $P < 0.05$, ** $P < 0.01$, *** $P < 0.001$.

4. While the trained immunity effect on tuft cells seems to be robust, an important question is whether this effect depends on other cell types in the intestine. Can the authors speculate if either local macrophages or lymphocytes may impact this training effect on tuft cells?

Response: Firstly, we appreciate the reviewer's suggestion and acknowledge that this trained immunity effect mainly depends on tuft cells and other cell types may affect this training effect weakly on tuft cells in the intestine. Therefore, we added a part of local macrophages or lymphocytes may impact this training effect on tuft cells in the discussion section in the revised manuscript [Lines:425-444], which was shown as below:

Although the trained immunity effect of the tuft cells seems to be robust in our study, the potential training effect of IL-25 on other intestinal innate immune cells, for instance, macrophages or lymphocytes, should not be ruled out. Macrophages have the ability to retain a long-lasting memory induced by previous microbial encounters, known as trained innate immunity. This enhanced feature of trained immune macrophages enables them to generate a more robust transcriptional response that is qualitatively and quantitatively distinct from that of untrained cells (PMID: 31530089). Available evidence showed that macrophages can be trained by LPS, β -glucan, or microbial moieties through long-term exposure (PMID: 27102489). IL-25 had been identified as an inducer for M2 macrophage polarization (PMID:28194019). M2 macrophage polarization can protect host from viral infection, including CVB3 (PMID: 28041873). However, it remains

unclear whether IL-25 plays a role in the training of macrophages. For lymphocytes, it has been reported that trained immunity in NK cells is important for its non-specific protection against infection. *Bacillus Calmette-Guérin* (PMID: 25451159), IL-12, IL-15, IL-18 (PMID: 35783158), etc., have been reported to have the capacity of inducing NK cell trained immunity. ILCs are recently discovered innate immune cells that can be trained. Until now, ILC1 can be trained by haptens and MCMV-m12; ILC2 can be trained by ASP, PAP, IL-33, and *S.venezuelensis*; ILC3 can be trained by *C.rodentium* (PMID: 36466877). IL-25 is important for ILCs activation, however, the role of IL-25 in lymphocytes is unknown. In short, IL-25, as an innate immune molecule, may also induce trained immunity in multiple innate immune cells, therefore, more research needs to be done in the future.

Referee #2 (Comments on Novelty/Model System for Author):

In this manuscript entitled "Trained Immunity of Intestinal Tuft Cells during Infancy Enhances Host Defense against Enteroviral Infections in Mice", Chen et al. describe that intestinal tuft cells expanded during EV71 infection. This expansion was associated with increased levels of IL-25. Likewise, recombinant IL-25 stimulation also led to tuft cell expansion. Tuft cell numbers decreased back to baseline after cessation of the infection, however, upon a secondary infection, they expanded strongly and quickly, and enhanced antiviral gene expression. The authors used both mouse *in vivo* and cell line models to functionally and mechanistically address the main questions of why infants are highly susceptible to EV71, the role of tuft cells in immunity to EV71, and whether tuft cells can acquire innate memory. Both the functional and mechanistic studies are carried out well, however the functional part of the study lacks key experiments. Moreover, in the current version of the manuscript, the authors do not provide an explanation for the maintenance and importance of tuft cells in memory immunity against EV71.

Response: We sincerely thank reviewer #2 for the positive feedback and the valuable comments. We have tried to consider all comments revised the manuscript based on the comments. In this revision, we performed additional experiments to explain the maintenance and importance of tufts in the memory immunity against EV71.

Major comments:

1. The authors aim to address the important question why infants are highly susceptible to EV71 infection, while older children and adults are not. Their hypothesis is that the tuft cells of older children and adults are trained through infection at infant age and therefore provide increased and faster antiviral immunity upon re-infection. This hypothesis is interesting. However, the study lacks basic experiments that provide the rationale for this research. Such as: EV71 infection in infant vs. adult wildtype mice. EV71 infection in adult mice that have or have not been infected/rmIL-25-treated during infancy. It is not clear, why the initial experiment has been conducted in Typ I/II IFN KO-mice, not WT mice at the age of 2 weeks.

Response: Thanks for your valuable comments.

(1) Based on your comments, we realized that this study does not fully explain the child and adult susceptibility to EV71. Therefore, we rewritten the manuscript, including abstract, introduction, results and discussion sections to re-organize the logic of this current study. In the newest manuscript, we primarily focus on EV71 infection induced tuft cell expansion via IL-25

production; subsequently, tuft cells can be trained by rIL-25; and then, IL-25-trained tuft cell support more IL-25 production to fight against enteroviruses, including EV71, CVA16 and CVB3; finally, we discuss the possible role of IL-25-trained tuft cells in the differences between child and adult susceptibility to EV71 [Lines:84-463].

(2) According to our previous study, EV71-Brcr strain will be completely cleared in WT mice on day 5-7 after peaking on day 3 (PMID: 38243751). Three days are too short to observe the EV71 clearance mediated by IL-25-trained tuft cells due to rmIL-25 induced tuft-cell expansion peak on day 7 after rmIL-25 administration. AG6 mouse (type I IFN receptor deficiency) has been reported as a susceptibility model for multiple viruses (ZIKA, DENV, etc), and it allows EV71-Brcr strain to replicate *in vivo* for 7 days reported by us (PMID: 37832655), which can also rule out the crosstalk from the IFN. Therefore, we initially performed the related experiments in AG6 mice. To illustrate the reasons why AG6 mice were chosen for experiments in our study, we added some sentences in Discussion section [Lines: 408-414], shown as below:

Our previous study showed that EV71-Brcr strain will be completely cleared in WT mice on day 5-7 after peaking on day 3 (PMID: 38243751). AG6 mouse (type I IFN receptor deficiency) has been reported as a susceptibility model for multiple virus, including ZIKA virus, dengue virus etc., which allows EV71-Brcr strain to replicate *in vivo* for 7 days reported by us (PMID: 37832655). Therefore, we initially chose AG6 mouse to investigate the relationship between EV71 infection and intestinal tuft cells expansion.

(3) Furthermore, we also added sentences in RESULTS 2.1 section [Lines: 161-166] to clarify that the experiments about enteroviruses-induced tuft cell expansion were also reproducible in *WT* mice. Because we have reported that enteroviruses, including EV71, CVA16, CVB3, and CVB4, could induce intestinal tuft cells expansion in enteroviral-infected C57BL/6J mouse model, which implied that this phenomenon was not specific to AG6 mouse.

2. In Fig 2 and 4, the authors elegantly describe how EV71 infection, IL-25 secretion, transient expansion of tuft cells, and increased protection against EV71 infection are linked. The ultimate experiment to prove this link would however be the infection of IL-25^{-/-} mice with EV71 and subsequent reinfection, measurement of disease score and survival. Studies in (immortalized) cell lines are somehow confusing in this context. In fact, they open up more questions on the source of IL-25 and antiviral molecules. What would be the relevance of these cell types in the intestines, particularly in the ileum? Focusing on *in vivo* mouse studies by IF or FACS will certainly be helpful to carefully delineate the real importance of tuft cells (and their potential memory function) in protection against EV71 infection. Moreover, the authors need to investigate whether increased cytokine and/or antiviral molecule levels (or their gene expression) is simply due to the expansion of the producing cells, or due to enhanced production capabilities of each single cell. This should be investigated through staining in isolated tuft cells from mice as well as upon *in vitro* re-stimulation.

Response: Thanks for your positive points about the results based on the Fig 2 and Fig 4 in the manuscript. We are sorry for the confuse to you about the exploring the mechanism of IL-25 mediated antiviral effect *in vivo*. Actually, we could not evaluate the disease score and obtain the survival rate in IL-25^{-/-} and *WT* mice because EV71 could be cleared in IL-25^{-/-} and *WT* mice

during the late time point following EV71 infection. We measured an increased viral load in the intestine at 7 dpi as compared to *WT* mice (Fig.5 in the revised manuscript). Therefore, we re-grouped the images in this revision in order to describe our study more clearly. Based on the data *in vivo*, we found that EV71 significantly induced tuft cells expansion via inducing IL-25 upregulation, and further found that IL-25 was produced in tuft cell trained immunity. Previous studies have uncovered that tuft cells, an epithelial cell type, are the single source of IL-25 in the small intestine (PMID:29069535). And we found that IL-25-trained tuft cells support more IL-25 production during EV71 infection as compared with non-trained tuft cells. We speculated that IL-25 may play an important role in IL-25-trained tuft cell mediated antiviral effects. Therefore, to confirmed the antiviral effects, we performed *in vitro* and *in vivo* experiments. We initially assessed the antiviral effects *in vitro*, and found that IL-25 over-expression or rIL-25 treatment both inhibited enteroviral replication in HeLa cells and Caco-2 cells (Fig. 4). And then we found that the anti-enteroviral effect was abolished in the IL-25-trained tuft cells *Il-25^{-/-}* mouse as compared with IL-25-trained tuft cells *WT* mouse (Fig. 5A-D and Fig.2L-N). Above evidence implied that IL-25 as an antiviral effector inhibited enteroviral replication *in vivo* and *in vitro*.

In the experiments of IL-25 antiviral effect assessment *in vitro*, HeLa cells and Caco-2 cells were chosen by us. HeLa cells line was commonly used in antiviral research since HeLa cells have the properties including convenient culture and easy accessibility (PMID:4306196). And Caco-2 cells are human clonal colon adenocarcinoma cells that closely resemble differentiated small intestinal epithelial cells in both structure and function, and are commonly utilized in research settings to conduct experiments that simulate the behavior of intestinal epithelial cells *in vivo* (PMID:28362399; PMID:37290501; PMID:22384261). Additionally, IL-25 expression and secretion can be detected when Caco-2 cells were infected with EV71. To a certain extent, it is possible to simulate tuft cells being infected by EV71, and EV71-mediated IL-25 training promotes tuft cells expansion and secrete IL-25 *in vivo*. The corresponding answer of Caco-2 cells was given in the 7th question from the reviewer 3#. Therefore, HeLa cells and Caco-2 cells were used for assessing anti-enteroviral effects mediated by IL-25 *in vitro* in this study. The above sentences have added in the section of RESULTS 2.4 in the revised manuscript. [Lines:216-224]

Furthermore, in our current study, this is a very valuable suggestion from the reviewer to distinguish the increased cytokine IL-25 derived from the expansion of the producing cells not an enhanced capability of each single cell. Due to the rarity of the tuft cells and technical difficulty in isolating and culturing them, we used DCLK1-IN-1, a selective inhibitor of doublecortin like kinase 1 to inhibit the expansion of tuft cell *in vivo* and investigated whether the ability of tuft cell expansion was abrogated to clear EV71 *in vivo*. Unfortunately, we found that DCLK1-IN-1 treatment reduced the viral load in ileum, which was shown as below:

A previous study has reported that DCLK1-IN-1 exhibited the anti-SARS-CoV-2 effect via blocking pro-inflammatory caspase-1/interleukin-1 β signaling in infected cells *in vitro* model (PMID: 35943255), but not tuft cells. DCLK1-IN-1 was a small molecular targeting of doublecortin like kinase 1 (PMID: 34545159). Based on the current result, we speculated that DCLK1-IN-1 exhibited anti-EV71 effect via suppressing doublecortin like kinase 1 not depending on the -tuft cell-mediated the IL-25/SAT1 signaling pathways.

The reviewer's concern is well taken, we will try to overcome technical challenges, in particular, to develop culture conditions for tuft cells, which will in the long run facilitate our mechanistic study on the cells.

Nevertheless, we added a paragraph to declare the limitation in the section of DISCUSSION [Lines:445-448], which was shown as below:

However, there are limitations in this study. Based on the current evidence, we have not been able to demonstrate that the increase in IL-25 and SAT1 is solely due to the expansion of IL-25-trained tuft cells. Therefore, further studies are needed in the future.

3. (1) Tuft cells are rare cells in the intestines. The authors should elaborate on this in the introduction, such as their frequency in the different parts of the intestines as well as their development and function. (2) Specifically, as the half-life of most tuft cells is only 4-5 days, how do the authors propose that training is maintained for >4 weeks in mice? (3) Is it through training in their progenitor cells or are long-lived tuft cells reprogrammed that specifically drive the secondary response? These questions need to be addressed using tuft cell-specific knockout or reporter mice (see e.g., Chandrakesan et al., *Oncotarget*, 2015, PMID:26362399).

Response: We thank the reviewer for the suggestion and for raising an important question. We have rewritten the INTRODUCTION sections according to your suggestion. Firstly, we provided details about tuft cells introduction, such as their frequency in the different parts of the intestines as well as their development and function [Lines:101-115], which was shown as below:

Tuft cells had been identified as innate immune cells in intestinal and lung tissues. Tuft cells have been reported to undergo rapid and transient hyperplasia after stimulation, who played an important role in the clearance of parasites and bacteria. However, it is not well understood whether tuft cells can be trained. Actually, tuft cells comprise approximately 0.5% of small

intestinal epithelial cells and colonic epithelial cells, but tuft cells are more prevalent in the distal than proximal small intestine, especially in the ileum. Tuft cells have been reported as originating from Lgr5⁺ stem cells, and doublecortin-like kinase 1 (DCLK1) or/and transient receptor potential melastatin 5 channel (TRPM5) are identified as tuft cell biomarker (PMID:26362399; PMID:30478383; PMID:37531421). Recent studies demonstrated that intestinal tuft cells expanded rapidly in response to enteric pathogens infection, which was associated with infection induced interleukine-25 (IL-25) expression (PMID: 38243751). Interestingly, IL-25 could be upregulated by multiple pathogens infection or even by alterations in the abundance of intestinal commensal bacteria (PMID:34704776; PMID:37040489; PMID:18762578). Therefore, IL-25 has been identified as an inducer of tuft cell expansion during microbes infection/colonization. However, the role of IL-25 in intestinal tuft cells training to fight against infections is intriguing.

In our present study, we observed that the number of tuft cell expansion in intestine tissues was increased induced by EV71 and peak at 7 dpi but it declined to a baseline at 14 dpi, which is consistent with other reports that the lifespan of intestinal tuft cells is only 4-5 days. In fact, our present study emphasizes that IL-25 or EV71-trained tuft cells acquire an immune memory that can last for 3~4 weeks, rather than maintaining intestinal tuft cell numbers due to the number of tuft cells have returned to the normal levels at 6 weeks old before EV71 infection (Fig. 3B~E). Chandrakesan et al., reported that Dclk1⁺ small intestinal epithelial tuft cell displays the hallmarks of self-renewal and quiescence (Chandrakesan et al., *Oncotarget*, 2015; PMID: 26362399). Actually, Dclk1⁺ cells may maintain quiescence, pluripotency, and metabolic activity for survival/longevity (PMID: 26362399). We speculate that the virus-infected or rIL-25-treated tuft cells differentiated or re-programmed to gain trained phenotype and may become quiescent, which respond to later stimulation at a larger scale. At present, we can not identify IL-25 trained tuft cells expansion derived from the same tuft cells or the re-programmed tuft cells in the current study. To demonstrate, we need to overcome the technical challenges in isolating sufficient number of tuft and developing ways of maintaining them in culture. Thus, it was deserved the further study to explore the original tuft cell.

Tuft cell expansion is associated with IL-25 stimulation. Fortunately, it is well established that IL-25-derived tuft cells expansion is dependent on IL-25-mediated ILC2s activation (PMID:26625736). IL-25-stimulated ILC2s secreting IL-13 that acts on epithelial crypt progenitors to promote differentiation of tuft cells, leading to increased tuft cell frequencies (PMID:26625736). Therefore, we speculated that IL-25-trained tuft cell may associated with ILC2s, but not directly with progenitor cells of tuft cells or tuft cell that specifically drive the secondary response.

Finally, we added these sentences to the paragraph in DISCUSSION section [Lines:445-457], which was shown as below:

However, there are limitations in this study. Based on the current evidence, we have not been able to demonstrate that the increased expression in IL-25 and SAT1 is solely due to the expansion of IL-25-trained tuft cells. Furthermore, it remains unclear whether IL-25-induced tuft cell training occurs directly through the tuft cell or through its progenitor cell. It is well established that IL-25-derived tuft cells expansion is dependent on IL-25-mediated ILC2s activation.

IL-25-stimulated ILC2s secreting IL-13 that acts on epithelial crypt progenitors to promote tuft cells differentiation, leading to increased frequencies. Therefore, we speculated that IL-25-trained tuft cell may be associated with ILC2s, the progenitor cells of tuft cells, but not directly with tuft cell that specifically drive the secondary response. Nevertheless, further exploration of the specific mechanisms of IL-25-derived tuft cell training is still needed in the future.

Minor Comments:

1. Please proofread manuscript by a native English speaker. While the English is ok throughout most of the text, the choice of especially verbs is uncommon and confusing in multiple places.

Response: Thanks for your careful reading and we have checked the full text to correct the grammatical error in our draft.

2. Introduction, ll. 113: More cell types have been shown to acquire immune training. Furthermore, differentiate between listing the signaling pathway (cytokines) vs the inducer (e.g., pathogen) of immune training.

Response: Thanks for your comments. We have added the following sentences to the INTRODUCTION section [Lines:89-96]: “Until now, increasing cell types have been shown to acquire immune training. Innate immune cells can be trained by specific cytokines or pathogens, which in turn trigger a cascade of increasingly robust events, such as enhanced cytokines production or accelerated expansion of trained cells. For instance, natural killer (NK) cells, group 2 innate lymphoid cells (ILC2) and group 3 innate lymphoid cells (ILC3) have been shown to exhibit the trained innate immune phenotypes to fight against pathogen infections, which are dependent on immune training with specific cytokines (IL-12 and IL-18, IL-33) and pathogens (C. rodentium), respectively” .

3. Why has an intracranial infection of 2-week old AG6 mice been used for EV71 and CVA16 or is this a misspelling in the Material and Methods section?

Response: Thanks for pointing out the error. It should be intraperitoneal injection instead of intracranial infection. We corrected the mistake in the Material and Methods section in revised manuscript [Lines:510].

4. What route of injection was used for rmIL-25 in C57BL/6 mice?

Response: C57BL/6 mice were treated with rmIL-25 by intraperitoneal injection and we added the respective information in the Material and Methods section accordingly in the revised manuscript. [Lines:555-556].

5. In models, please indicate age and sex of the mice.

Response: We added the information about age and sex of the mice in Fig. 1A, Fig. 2G and L, Fig. 3A, 3F, 3K, Fig. 5A, Fig. 7A in the revised manuscript.

6. Fig 3 has been mislabelled in the text as Fig 4.

Response: We have re-arranged the figures, and carefully checked all the label in the figures to avoid the mislabeling.

7. Fig 1G+H, what is the time point of this cytokine measurement?

Response: For Fig 1G-H, IL-25 was measured by ELISA at 7 days post-infection (dpi). We have annotated the 7 dpi time point in Fig. 2A-B (originally presented as Fig. 1G-H) in the updated manuscript.

8. Fig 1A and I, it is not clear from the models if the infection/recombinant cytokine injections were only once (on d 0) or on multiple days?

Response: Thanks for your suggestions. We added the key information to present the models in Fig. 1A and Fig. 2G (originally presented as Fig. 1I). In Fig. 1A, the mouse was infected EV71 1.2×10^6 PFU/mouse by i.p. on day 0, and the intestine tissues were collected on 3,5,7,14 dpi, respectively. In Fig. 2G (originally presented as Fig. 1I), the mouse was stimulated rIL-25 on day 1, 3, 5 by i.p., and the ileum tissues were collected on day 7. We have redrawn the schematic diagram and provided these details in the revised manuscript.

9. Fig 1 J, K, L, M do not show different time points as indicated in Fig 1I. Please note, what time point is shown for these analyses.

Response: Thanks for your suggestions. The time point is at 7d for the harvest of ileum tissue. In this version, we labeled the time points of these experiments in Fig 2H-K (originally presented as Fig. 1J-M) in the revised manuscript.

10. Fig 2 M-T: It is not clear, what the "fold change" of mRNA expression relates to. In general, it is more informative to provide the total values instead of the fold changes.

Response: Thanks for your suggestion. We have corrected all the related descriptions throughout the text.

Referee #3 (Comments on Novelty/Model System for Author):

Despite the chosen grades, there are major limitations to this manuscript as indicates below.

1. The manuscript text and structure as well as figure illustration is hard to follow, the language should be revised.

Response: Thanks for your comments. According to your comments, we re-organized the logic of this study and rewritten the manuscript, including abstract, introduction, results and discussion sections. Furthermore, we also re-group the figures. In the revised manuscript, we initially focused on EV71 infection could induce tuft cell expansion (Fig. 1), and then we found that EV71-induced IL-25 was involved in EV71-induced tuft cell expansion in the ileum tissues (Fig. 2). Subsequently, we investigated whether tuft cells could be trained by IL-25 and found that tuft cells could be trained IL-25 and IL-25-trained tuft cell exhibited a stronger anti-enteroviral response (Fig. 3). A previous study have uncovered that tuft cells, an epithelial cell type, were the single source of IL-25 in the small intestine (PMID: 29069535). And we found that IL-25-trained tuft cell supported more IL-25 production to fight against enteroviruses, including EV71, CVA16, and CVB3, therefore, IL-25 was identified as an effector for IL-25-trained tuft cell-mediated anti-enteroviral effects in our study (Fig. 4 and Fig. 5). Finally, the data showed that

SAT1-mediated polyamine depletion was participated in IL-25-induced anti-enteroviral effects (Fig. 6, Fig. 7 and Fig. 8).

Furthermore, we asked our native English speaking friends to help us promote the quality of the manuscript.

2. Important experimental information on virus strains and tools is lacking.

Response: We have added the viral strains and tools in the revised manuscript [Lines: 481-486].

Referee #3 (Remarks for Author):

Chen et al. use a type I and II receptor deficient mouse model to study the effect of IL-25 on enteric enterovirus 71 (EV71) replication. They show that (i) IL-25 is sufficient to increase Tuft cells and required for EV71 induced Tuft cell increase at 2 weeks post infection; (ii) rIL-25 treatment or administration of a non-lethal EV71 dose leads on a transient Tuft cell number increase (peak at 2 week, baseline at 6 weeks). Infection 6 weeks after infant (2 week-old mice) rIL-25 treatment or EV71 infection leads to enhanced IL-25 levels and Tuft cell numbers (over non IL-25/EV71 infected controls) and a reduced EV organ load; (iii) viral protein expression in epithelial cell lines induces IL25 expression and secretion; (iv) IL-25 overexpression/rIL25 administration/transgene expression of IL-25 using a CMV system suppresses viral protein (VP1) expression and EV71 infection in IL-25 KO mice leads to higher virus titers and no tuft cell expansion; IL-25 transfected HeLa cells upregulate SAT1 (among other genes) and IL-25 treatment of ileal tissue increases SAT1 expression 2-fold, knock-down of SAT1 leads to increased EV71 replication in Caco2 cells, SAT1 overexpression leads to decreased viral gene expression in HeLa cells. SAT1 knockdown/SAT1 inhibitor abrogate the protective IL-25 effect on viral replication; (v) SAT1 overexpression leads to polyamine depletion (known, polyamines required for RNA virus replication) and (vi) diminazene aceturate (DA, a SAT1 inhibitor) administration abolishes the trained effect of early life rIL-25 treatment in vivo.

Generally, the data shown are timely, medically relevant and interesting. However, there are a number of issues that make this manuscript not suitable for publication in the present form.

Response: We thank the reviewer for the insightful and valuable comments, and we have addressed all the raised concerns and suggestions. As a result, the quality of the manuscript has been substantially improved. Please find our detailed point-by-point response listed below.

Major points:

1. Please define the target cell for EV71 in the AG6 model, are tuft cells infected?

Response: To address the question, we conducted an additional experiment. 2 weeks old AG6 male mice were infected with EV71 for 7 dpi, and ileum tissues were collected. We observed that EV71 could infect intestinal epithelial cells and tuft cells as VP1 (green) was detected in intestinal epithelial cells and DCLK1+ cells. The result was added to **Figure EV1**, and the data was shown as below:

Figure EV1 AG6 male mice were infected with EV71 for 7 dpi, and ileum tissues were collected. Immunofluorescence assays were used to detect EV71-viral protein VP1 expression (Green) and DCLK1 expression (Red).

2. Please include the statistical test used for each panel (legend text). If parametric tests are used test for normal distribution.

Response: Thanks for your valuable comments. According to you and Reviewer#1 comments, we re-analyzed all the data used in this study, and sought the help of statistical professionals to standardize the use of statistical methods. In this revision, we mentioned the primary statistical methods used for each set of data in the figure legend and revised the description of statistical analysis in METHODS section [Lines:654-664], which is shown as below:

Statistical analyses were performed using GraphPad Prism 9.0 software (GraphPad, San Diego, CA, USA). All data were initially determined to be normally distributed or not by the Shapiro-Wilk test. For data that conforms to a normal distribution, comparisons between two groups were analyzed using two-tailed Student's t-test. One-way ANOVA followed by Dunnett's-t test was used to compare more than two data groups, which were run only if F-test achieved $P > 0.05$, and there was no significant variance in homogeneity. Nonparametric tests were used for data sets that did not conform to a normal distribution. P values were provided as * $P < 0.05$, ** $P < 0.01$, *** $P < 0.001$.

3. Please discuss the source of IL-25 under *in vivo* conditions.

Response: We have discussed the source of IL-25 from two aspects. (1) we included a previous study showing that tuft cells, an epithelial cell type, are the single source of IL-25 in the small intestine (PMID: 29069535) (Lines:355-356); (2) we also discussed the inducer of IL-25 *in vivo*

conditions. We emphasized the important role of intestinal commensal bacteria, pathogenic infection, food allergens, and contaminant exposure in the induction of IL-25 in the intestinal (Lines:323-347).

4. Please discuss the AG6 mouse model that lack IFN type I and II receptor expression (in the material and methods section called alpha, beta and gamma receptor, give a more detailed description). How could the lack of IFN I and II interfere with IL-25 and tuft cell numbers? Also type III IFN is still present in these mice and should in the epithelium confer resistance, discuss why the model is suitable. Notably SAT1 was reported to be IFN-induced (Mounce et al., Cell Host Microbe, 2016).

Response: To address your concern, we added the statement as following [Lines:529-530]:
AG6 mice (*IFN - α / β and γ receptor* deficient) with the C57BL/6 genetic background were provided by Prof. Qibin Leng of Guangxi Medical University.

As for the suitability of AG6 model, we added the following description to the section of DISCUSSION [Lines:404-423]:

Interferons (IFNs) are common and important cytokines that inhibited viral infection. AG6 mouse has been reported as a susceptible model for multiple virus, including ZIKA virus, dengue virus etc. Our previous study showed that EV71-Brcr strain was completely cleared in WT mice on day 5-7 after peaking on day 3 (PMID: 38243751), while AG6 mouse allows EV71-Brcr strain to replicate *in vivo* for 7 days (PMID: 37832655). Secondly, AG6 mice have the potential to amplify IL-25-tuft cell signaling. IL-25-induced tuft cell expansion is dependent on IL-25-mediated ILC2 proliferation and ILC2-associated IL-13 production. Han M et al., found that in immature mice, IFN- γ inhibited ILC2 proliferation and IL-13 production (PMID: 27679954). Moreover, multiple studies showed that IFNs as negative regulators of ILC2s (PMID: 37414870; PMID: 27027961; PMID: 30294963), which suggested that IFNs could be inhibitory factors for tuft cell expansion. The phenomenon of tuft cell expansion induced by EV71 infection or IL-25 stimulation is not exclusive to AG6 mice, as we have also observed the same response in EV71-infected or IL-25-stimulated C57BL/6J (*WT*) mice (PMID: 38243751). In conclusion, AG6 mice present a suitable model for exploring the interplay between viral infections and tuft cell responses.

Finally, we also learn from Mounce et al., that Type I interferon induces SAT1, a catabolic enzyme in the host polyamine pathway (Mounce et al., Cell Host Microbe, 2016). And we added the relevant discussion to the section of DISCUSSION [Lines:396-402] as following:

Mounce et al., reported IFN- β as an inducer of SAT1 (PMID: 27427208), while we reported IL-25 as an another inducer of SAT1. A previous study showed that p53, a transcription factor, promoted SAT1 expression, and further suggested SAT1 was a direct p53 target (PMID: 27698118). Interestingly, p53 also plays a critical role in the activation of the tuft cell-IL-25-type 2 innate lymphoid cell circuit (PMID: 34099671), which implies that IL-25 exhibits anti-enteroviral effect through upregulating SAT1 expression.

5. The definition of the time window (< and > 2 weeks) that facilitates the described trained immunity phenotype would add to the manuscript.

Response: We have revised the description about trained immunity phenotype [Lines: 204-207 and Lines316-317].

6. Improvement of the text by a native English-speaking person would significantly improve the clarity of the manuscript.

Response: We asked our native English speaking friends to help us to improve the quality of the manuscript.

7. Does IL-25 overexpression in intestinal epithelial cell lines leads to cytokine secretion in the supernatant? Do epithelial cells in vivo (cell lines in this study) express the IL-25 receptor?

Response: In order to address your questions, we detected the IL-25 cytokine in the supernatant in IL-25 overexpression in Caco-2 cells by ELISA assay. The results were shown as following. IL-25 overexpression in Caco-2 cells led to release IL-25 induced by EV71 infection. (2) As previous studies have demonstrated that tuft cells expressed IL-25 receptors including IL-17RA (PMID:35081371) and IL-17RB (PMID:31182574, PMID:29144463, PMID:36989894); Moreover, in order to address your questions, we analyzed the expression level of IL-17RA or/and IL-17RB in Caco-2 cells in the presence or in the absence of EV71 infection in our current study. We found that IL-17RB not IL-17RA was induced in Caco-2 cells following EV71 infection. We provided the results as follows:

Caco-2 cells overexpression IL-25 and the control group Caco-2 cells were both infected with EV71 for 24 h and ELISA assay was used to detect the cytokine IL-25 level in the cell supernatant. We measured an increased level of IL-25 in IL-25 overexpression Caco-2 cells.

Caco-2 cells were infected with EV71 at an MOI=3 for 24 h and IL-17RB, VP1 and GAPDH protein expression were detected by western blotting assay.

8. Fig. 3D is unclear (in legend panel D is described to show an IF) how can IL-25 and VP1 be shown on one single membrane?

Response: We thank the reviewer for pointing out the error and apologized for the mistake. In fact, panel F showed an IF result. The question that IL-25 and VP1 have been shown on one single membrane because the membrane was incubated with IL-25 antibody and it was scanned in Odyssey Infrared Imager. After then, the membrane was incubated with VP1 antibody and it was also scanned in Odyssey Infrared Imager. In order to avoid the confusion, we have replaced the images in the Fig. 2D (originally presented as Fig. 3D) manuscript with clearer ones, which is shown as below:

Fig. 2D: Caco-2 cells were infected with EV71 at different MOIs for 24 h and IL-25, VP1 and actin protein expression were detected by western blotting assay.

9. What was the rationale to select SAT1 for further studies?

Response: We have provided the reason why SAT1 was chosen for further studies. The relevant descriptions have added to RESULT 2.6 section [Lines: 255-259] as following:

To explore the potential mechanism of IL-25-mediated antiviral effects, we established *IL-25^{+/+}* HeLa cell line via lentiviral transfection assay. RNA sequencing (RNA-seq) was conducted to analyze the differential expression of transcriptional genes in *WT* and *IL-25^{+/+}* HeLa cells. SAT1 (spermidine/spermine acetyl-transferase enzyme) was chosen as a target gene for further study

since its mRNA was significantly increased by $\log_2\text{Fc}>2$ in *IL-25*^{+/+} HeLa cells (Fig. 6A). SAT1 is the rate-limiting enzyme controlling the first intracellular pathway of polyamine catabolism, and high level SAT1 expression leads to an overall depletion of polyamines intracellular (PMID: 27698118). Excitingly, the polyamines are essential for the replication of viruses (PMID: 28904024). Therefore, we focused on the roles of SAT1 expression in IL-25-mediated anti-enteroviral effects.

10. IL25 blockade was shown to augment antiviral immunity against (RNA) respiratory viruses (Williams et al., 2022, Commun Biol), Please discuss.

Response: We have added the following paragraph to the DISCUSSION section [Lines:339-347]: Teresa C Williams et al. found that exogenous IL-25 treatment increased the rhinovirus (RV) load, a common respiratory virus (PMID: 35508632), which was different from our findings. The discrepancy may be caused by animal models used. They found that anti-IL-25 monoclonal antibody (LNR125) treatment increased the RV load in an asthma model established by BALB/c, while they did not observe any changes in coronavirus 229E and RV viral load with LNR125 treatment in bronchial epithelial cells (BECs), which implied that ovalbumin established asthma BALB/c mouse model led to a different role for IL-25 in viral infections. Furthermore, viral specificity should not be ignored due to RV has a robust association with asthma, and IL-25 may support RV replication.

11. Please better explain the polyamine red assay. Is polyamine red secreted to allow OD measurements (not explained in M&M section).

Response: We have added detailed explanation about polyamine red assay to the METHODS section [Lines:633-641] as below:

The polyamine red assay was performed by using the total Polyamine Assay kit (PolyamineRED, Intracellular Polyamine Detection Reagent, Cat#: FDV-0020). PolyamineRED can permeate cellular membrane and couples TAMRA to polyamines inside the cell. The unreacted PolyamineRED is removed by washing with PBS and total polyamines can be semi-quantified by intracellular fluorescence intensity by fluorescence reader. The distribution of intracellular polyamines can be observed by a ZEISS microscope. The relative fluorescence intensities were analyzed by ImageJ software.

12. The quantification of Tuft cells is unclear? L521: How many sections, fields of view etc. were analyzed?

Response: We have provided the information about the the quantification of tuft cells in the section of Materials and Methods [Lines:627-630] as following: Graph showed the distribution of Dclk1⁺ tuft cells in ileum tissues in naive and EV71-infected mice on day 3, 5, 7 and 14 post infection. Tuft cells were counted in the crypt and villus compartments of n=50 crypt-villus units per mouse with 3-5 mice per treatment. Means of villus/crypt ratio of tuft cell numbers are shown. In order to provide the convenience to the readers, we have added the respective information in figure legend 1E in the revised manuscript [Lines:844-846].

13. Please better describe the virus strains used. The manuscript indicates the EV71 BrCr and CVA16 strains but which strain has been used for which panel is not indicated. The reference

Chen et al. indicated in the manuscript does not give any further information. The reference Fu et al. refers to a EV71 Fuyang0805 strain that is not used in this study. Please clarify.

Response: Thank you for your pointing out our oversight. We have added the viral strain information accordingly in the revised manuscript [Lines:480-485]. Enterovirus A71 (EV71) (BrCr strain) was a gift from Prof. Bin Wu, Jiangsu Provincial Center of Disease Control. Information has been added to the reference Chen et al. And the reference Fu et al., was removed in the revised manuscript.

Minor points:

1. Please discuss the effect of cGas which as been shown to be controlled by polyamine metabolism (Zhao et al., Immunity, 2023).

Response: This is a very valuable suggestion. We believed that the reviewer possesses the profound knowledge in the field of polyamine metabolism. We have cited the reference (Zhao et al., Immunity, 2023) and added a statement of cGAS induced by polyamine metabolism in the discussion section in the revised manuscript. [Lines:388-391]

2. Figure 1 C, please quantify villus lengths.

Response: In order to address your questions, we quantify villus lengths in Fig. 1C in revised manuscript.[Lines:154-155]

3. Please add bars in all figure panels.

Response: We added all bars in all figure panels according to your suggestions in revised manuscript.

4. Please comment on the role of ILC2s which respond to IL-25 and may be involved in the induction of tuft cell hyperplasia.

Response : We have added the related discussion in DISCUSSION section [Lines:449-454] as below:

Fortunately, it is well established that IL-25-derived tuft cells expansion is dependent on IL-25-mediated ILC2s activation (PMID:26675736). IL-25-stimulated ILC2s secreting IL-13 that acts on epithelial crypt progenitors to promote the differentiation of tuft cells, leading to an elevated tuft cell frequencies (PMID:26675736). Therefore, we speculated that IL-25-trained tuft cell may be associated with ILC2s, but not directly with progenitor cells of tuft cells or tuft cell that specifically drive the secondary response.

5. Paragraph 2.3 refers to Fig. 3 but indicates Fig. 4.

Response: We have corrected the mistakes in the revised manuscript.

6. Fig. 1 I-O may be moved to the supplemental information.

Response: We re-grouped the figures in the revised manuscript. To make the logic more complete, we have moved the original version Fig. 1I-O to the new version Fig. 2H-K.

7. Fig. 4 N-T may be moved to the supplemental information as well.

Response: Based on your suggestions, we re-grouped the figures in the revised manuscript and moved the original version Fig. 4 N-T to the new version Fig. 5, since the IL-25-trained tuft cells expansion is an event that occurs *in vivo*.

8. Fig. 5 V-W, indicate what color indicates what.

Response: We have added the color indicator in Fig. 6P and Q (originally presented as Fig. 5V-W) in the revised manuscript.

9. Fig. 7 B, please provide body weight not as difference but as weight (g) per animals.

Response: We have provided the body weights (g) as shown in Fig.7B accordingly in the revised manuscript.

10. The color used could be harmonized to improve clarity.

Response: The reviewer's suggestion is well taken and we have done so accordingly in the revised manuscript.

15th Jul 2024

Dear Dr. Chen,

Thank you for the submission of your revised manuscript to EMBO Molecular Medicine. I am pleased to inform you that we will be able to accept your manuscript pending the following final amendments:

- 1) Authors: E-mail correspondence to Jing Wu could not be delivered. Please update the e-mail address of this author. Also, in the manuscript Devyan Chen is listed as a first author and in our system as the corresponding author. Please make sure that first and corresponding author match in the manuscript and in our submission system.
- 2) In the main manuscript file, please do the following:
 - Please address all comments suggested by our data editors listed below:
 - o Figure legends:
 1. Please note that the legends for figures 3c-d is not provided in the sequential manner (legend for figure 3d is provided before legend of figure 3c). This needs to be rectified.
 2. Please define the annotated p values *** as well as provide the exact p-values for the same in the legend of figure EV 2c; as appropriate.
 3. Please note that the exact p values are not provided in the legends of figures 1b-c, e, h; 2a-b, i-k, n; 3b-c, h-j, m-o; 4a-b, d-e, g-l; 5b-d; 6b-o; 7b-h; 8b-d.
 4. Please indicate the statistical test used for data analysis in the legend of figure EV 2c.
 5. Please note that in figures 5b-d; 6b, f-g, i, m; there is a mismatch between the annotated p values in the figure legend and the annotated p values in the figure file that should be corrected.
 6. Please note that the box plots need to be defined in terms of minima, maxima, centre, bounds of box and whiskers, and percentile in the legends of figures 4a-b.
 7. Please note that information related to n is missing in the legends of figures 2a-b, n; 3h; 4a-b, d-e, g-l; 6b-q; 8b-c; EV 2c.
 8. Please note that the error bars are not defined in the legends of figure EV 2c.
 9. Please note that for heatmap present in figure 6a a numbered scale bar is not provided. This needs to be rectified.
 10. Please note that the white arrows are not defined in the legend of figure EV 1. This needs to be rectified.
 - Add callouts for Fig 5C and Fig 8B.
 - Rename "Conflicts of interests" to "Disclosure and competing interests statement". We updated our journal's competing interests policy in January 2022 and request authors to consider both actual and perceived competing interests. Please review the policy <https://www.embopress.org/competing-interests> and update your competing interests if necessary.
 - Author contributions: Please remove it from the manuscript and specify author contributions in our submission system. CRediT has replaced the traditional author contributions section because it offers a systematic machine-readable author contributions format that allows for more effective research assessment. You are encouraged to use the free text boxes beneath each contributing author's name to add specific details on the author's contribution. More information is available in our guide to authors: <https://www.embopress.org/page/journal/17574684/authorguide#authorshipguidelines>
 - Please include structured Methods section that includes a Reagents and Tools Table followed by a Methods and Protocols section. More information on how to adhere to this format as well as downloadable templates (.docx) for the Reagents and Tools Table can be found in our author guidelines: <https://www.embopress.org/page/journal/17574684/authorguide#structuredmethods> An example of a paper with Structured Methods can be found here: <https://www.embopress.org/doi/full/10.1038/s44320-024-00037-6#sec-4>
 - In Methods, provide the antibody dilutions that were used for each antibody.
 - In Methods, a statistical paragraph should reflect all information that you have filled in the Authors Checklist, especially regarding randomization, blinding, replication.
 - Indicate in legends number and nature of replicates and exact p= values, not a range, along with the statistical test used. To keep the figures "clear" some authors found providing an Appendix table Sx with all exact p-values preferable. You are welcome to do this if you want to.
 - Correct the reference citation in the text and reference list. In the text a reference should be cited by author and year of publication. Include a space between a word and the opening parenthesis of the reference that follows. In the reference list, citations should be listed in alphabetical order. Where there are more than 10 authors on a paper, 10 will be listed, followed by "et al.". Also, please remove DOIs. Please check "Author Guidelines" for more information. <https://www.embopress.org/page/journal/17574684/authorguide#referencesformat>
 - Data availability: Please use the following format to report the accession number of your data:

[data type]: [full name of the resource] [accession number/identifier] [(doi or URL or identifiers.org/DATABASE:ACCESSION)]

Please check "Author Guidelines" for more information.

<https://www.embopress.org/page/journal/17574684/authorguide#availabilityofpublishedmaterial>

3) Tables and EV Figures: Please add Tables 1 and 2 with table legends to the manuscript text, after the main figures. The legends of Fig EV1-3 should be added to the manuscript text, after the tables and with the heading "Expanded View Figure Legends".

4) Funding: Please merge it with the Acknowledgments.

5) Synopsis:

- Synopsis text: Please remove it from the manuscript and upload it as a separate .doc file.

- Synopsis image: Please resize the image to 550 px-wide x (250-400)-px high and upload it as a high-resolution jpeg file.

6) For more information: This space should be used to list relevant web links for further consultation by our readers. Could you identify some relevant ones and provide such information as well? Some examples are patient associations, relevant databases, OMIM/proteins/genes links, author's websites, etc..

7) As part of the EMBO Publications transparent editorial process initiative (see our Editorial at

<http://embomolmed.embopress.org/content/2/9/329>), EMBO Molecular Medicine will publish online a Review Process File (RPF) to accompany accepted manuscripts. This file will be published in conjunction with your paper and will include the anonymous referee reports, your point-by-point response and all pertinent correspondence relating to the manuscript. Let us know whether you agree with the publication of the RPF and as here, if you want to remove or not any figures from it prior to publication. Please note that the Authors checklist will be published at the end of the RPF.

8) Please provide a point-by-point letter INCLUDING my comments as well as the reviewer's reports and your detailed responses (as Word file).

I look forward to reading a new revised version of your manuscript as soon as possible.

Yours sincerely,

Zeljko Durdevic

*** Instructions to submit your revised manuscript ***

1) a .docx formatted version of the manuscript text (including Figure legends and tables)

2) Separate figure files*

3) supplemental information as Expanded View and/or Appendix. Please carefully check the authors guidelines for formatting Expanded view and Appendix figures and tables at <https://www.embopress.org/page/journal/17574684/authorguide#expandedview>

4) a letter INCLUDING the reviewer's reports and your detailed responses to their comments (as Word file).

5) The paper explained: EMBO Molecular Medicine articles are accompanied by a summary of the articles to emphasize the major findings in the paper and their medical implications for the non-specialist reader. Please provide a draft summary of your article highlighting

This may be edited to ensure that readers understand the significance and context of the research.

Please refer to any of our published articles for an example.

6) For more information: There is space at the end of each article to list relevant web links for further consultation by our readers. Could you identify some relevant ones and provide such information as well? Some examples are patient associations, relevant databases, OMIM/proteins/genes links, author's websites, etc...

7) Author contributions: the contribution of every author must be detailed in a separate section.

8) EMBO Molecular Medicine now requires a complete author checklist

(<https://www.embopress.org/page/journal/17574684/authorguide>) to be submitted with all revised manuscripts. Please use the checklist as guideline for the sort of information we need WITHIN the manuscript. The checklist should only be filled with page numbers where the information can be found. This is particularly important for animal reporting, antibody dilutions (missing) and exact values and n that should be indicated instead of a range.

9) Every published paper now includes a 'Synopsis' to further enhance discoverability. Synopses are displayed on the journal webpage and are freely accessible to all readers. They include a short stand first (maximum of 300 characters, including space) as well as 2-5 one sentence bullet points that summarise the paper. Please write the bullet points to summarise the key NEW findings. They should be designed to be complementary to the abstract - i.e. not repeat the same text. We encourage inclusion of key acronyms and quantitative information (maximum of 30 words / bullet point). Please use the passive voice. Please attach these in a separate file or send them by email, we will incorporate them accordingly.

You are also welcome to suggest a striking image or visual abstract to illustrate your article. If you do please provide a jpeg file 550 px-wide x 300-600px high.

10) A Conflict of Interest statement should be provided in the main text

11) Please note that we now mandate that all corresponding authors list an ORCID digital identifier. This takes <90 seconds to complete. We encourage all authors to supply an ORCID identifier, which will be linked to their name for unambiguous name identification.

Currently, our records indicate that the ORCID for your account is 0000-0001-7351-4617.

Link Not Available

12) Include a Reagents and Tools Table as part of the Methods section, which can be downloaded from our author guidelines (<https://www.embopress.org/page/journal/17574684/authorguide#structuredmethods>)

Photos 400-800 DPI

*Additional important information regarding figures and illustrations can be found at

<https://bit.ly/EMBOPressFigurePreparationGuideline>. See also figure legend preparation guidelines:

<https://www.embopress.org/page/journal/17574684/authorguide#figureformat>

***** Reviewer's comments *****

Referee #1 (Comments on Novelty/Model System for Author):

Very good, relevant study.

Referee #1 (Remarks for Author):

The authors responded appropriately to my suggestions.

The authors addressed the editorial issues.

12th Aug 2024

Dear Dr. Wu,

We are pleased to inform you that your manuscript is accepted for publication and is now being sent to our publisher to be included in the next available issue of EMBO Molecular Medicine.
